# ACAD10 and ACAD11 enable mammalian 4-hydroxy acid lipid catabolism

Edrees H. Rashan[1,12], Abigail K. Bartlett[1,2,12], Daven B. Khana[3], Jingying Zhang[2,4], Raghav Jain [1], Gina Wade[1], Luciano A. Abriata [5], Andrew J. Smith [2], Zakery N. Baker [2], Taylor Cook[6], Alana Caldwell [1], Autumn R. Chevalier[1], Patrick Forny [2], Brian F. Pfleger [6], Matteo Dal Peraro [5], Peng Yuan [2,4], Daniel Amador-Noguez [3], Judith A. Simcox [1,7] ✉ & David J. Pagliarini [1,2,8,9,10,11] ✉

Fatty acid β-oxidation is a central catabolic pathway with broad health implications. However, various fatty acids, including 4-hydroxy acids (4-HAs), are largely incompatible with β-oxidation machinery before being modified. Here we reveal that two atypical acyl-CoA dehydrogenases, ACAD10 and ACAD11, drive 4-HA catabolism in mice. Unlike other ACADs, ACAD10 and ACAD11 feature kinase domains that phosphorylate the 4-hydroxy position as a requisite step in converting 4-hydroxyacyl-CoAs into conventional 2-enoyl-CoAs. Through cryo-electron microscopy and molecular modeling, we identified an atypical dehydrogenase binding pocket capable of accommodating this phosphorylated intermediate. We further show that ACAD10 is mitochondrial and necessary for catabolizing shorter-chain 4-HAs, whereas ACAD11 is peroxisomal and enables longer-chain 4-HA catabolism. Mice lacking ACAD11 accumulate 4-HAs in their plasma and females are susceptible to body weight and fat gain, concurrent with decreased adipocyte differentiation and adipokine expression. Collectively, we present that ACAD10 and ACAD11 are the primary gatekeepers of mammalian 4-HA catabolism.

The core enzymes of mitochondrial fatty acid β-oxidation (FAO) were discovered in the mid-20th century following the classic cell fractionation work of Claude, Lehninger and others[1–3]. In the ensuing decades, additional enzymes (for example, reductases and isomerases) required to process select unsaturated fatty acids before they could enter the FAO cycle were identified. Additionally, a second venue for FAO—the peroxisome—was revealed, along with enzymes in this organelle critical for the catabolism of branched and very-long-chain fatty acids[4,5]. To date, at least 17 proteins related to fatty acid catabolism have been directly linked to human genetic diseases and aberrant fatty acid metabolism is associated with many metabolic disorders, including obesity, diabetes, glioblastoma and chronic liver and kidney dysfunction[6–11].

Discoveries of new fatty acid species—driven by advances in analytical techniques, such as mass spectrometry (MS)[12]—are further expanding our understanding of lipid metabolism and highlighting the

[1]Department of Biochemistry, University of Wisconsin–Madison, Madison, WI, USA. [2]Department of Cell Biology and Physiology, Washington University School of Medicine, St. Louis, MO, USA. [3]Department of Microbiology, University of Wisconsin–Madison, Madison, WI, USA. [4]Department of Pharmacological Sciences, Icahn School of Medicine at Mount Sinai, New York, NY, USA. [5]Laboratory for Biomolecular Modeling and Protein Structure Core Facility, School of Life Sciences, École Polytechnique Fédérale de Lausanne and Swiss Institute of Bioinformatics, Lausanne, Switzerland. [6]Department of Chemical and Biological Engineering, University of Wisconsin–Madison, Madison, WI, USA. [7]Howard Hughes Medical Institute, University of Wisconsin–Madison, Madison, WI, USA. [8]Morgridge Institute for Research, Madison, WI, USA. [9]Department of Biochemistry and Molecular Biophysics, Washington University School of Medicine, St. Louis, MO, USA. [10]Department of Genetics, Washington University School of Medicine, St. Louis, MO, USA. [11]Howard Hughes Medical Institute, Washington University School of Medicine, St. Louis, MO, USA. [12]These authors contributed equally: Edrees H. Rashan, Abigail K. Bartlett. ✉e-mail: jsimcox@wisc.edu; pagliarini@wustl.edu

requirement for additional enzymes that enable their production and catabolism. A recent example is the family of branched fatty acid esters of hydroxy fatty acids (FAHFAs), which has been linked to beneficial effects on glycemia, insulin secretion and inflammation[13]. A second, less studied family is the 4-hydroxy acids (4-HAs), which comprises fatty acids with a hydroxy (−OH) group on the 4 (γ)-position. 4-HAs are produced by common lipid peroxidation, formed by enzymatic conversion or the catabolism of longer-chain hydroxy acids and can be ingested in the form of various naturally occurring lipids or certain drugs of abuse (for example, 4-hydroxybutyrate or 4-hydroxyvalerate (4-HV))[14–17]. Indeed, recent MS analyses identified a spectrum of 4-HAs and 4-HA precursors in human serum, yet few 4-HAs have been studied at any detail in mammalian systems[12].

## Results

### Atypical acyl-coenzyme A dehydrogenases 10 and 11 process 4-HA substrates in vitro and in cells

A central question in 4-HA biology is what enzymes catabolize these lipids following their acylation with acyl-coenzyme A (CoA). The presence of an −OH group on the γ-carbon likely reduces their compatibility with the established acyl-CoA dehydrogenases (ACADs) of FAO, which catalyze the α,β-dehydrogenation of acyl-CoA substrates to produce 2-enoyl-CoAs (Fig. 1a). Nonetheless, previous MS-based metabolic tracing analyses performed on murine livers perfused with various 4-HAs demonstrated that these lipids can be catabolized through two pathways: a minor pathway, involving a sequence of β-oxidation, α-oxidation and β-oxidation steps, and a major pathway—5–6-fold more active than the minor—involving a unique phosphorylated acyl-CoA intermediate (Fig. 1a). However, no mammalian enzymes required for this major pathway have been identified[18–21]. More recently, a similar pathway was observed for the breakdown of levulinic acid (Lva) in *Pseudomonas putida*[22]. Here, Lva is converted into 4-HV, a short-chain 4-HA, that then proceeds into FAO through a similar phosphorylated intermediate. The authors further identified the necessary '*lva* operon' for this pathway, which includes genes encoding a kinase-like protein (LvaA) and an ACAD family protein (LvaC) (Fig. 1b).

To identify potential mammalian orthologs of LvaA and LvaC, we performed a protein basic local alignment search tool (BLASTp) analysis. Surprisingly, the top hits for LvaA and LvaC were the same two proteins, acyl-CoA dehydrogenases 10 and 11 (ACAD10/11), two poorly characterized ACAD paralogs. LvaA exhibits high homology to N-terminal kinase domains found in ACAD10/11 but not in other members of the ACAD family (Fig. 1b and Extended Data Fig. 1a). These kinases possess all the core catalytic residues and subdomain motifs of the protein kinase-like (PKL) superfamily and show particular homology to the aminoglycoside kinase–phosphotransferase 3 (APH3) subgroup of eukaryotic-like kinases (Extended Data Fig. 5c)[23]. LvaC exhibits homology to the C-terminal ACAD domains of ACAD10/11. Among all mammalian ACADs, LvaC is most similar to ACAD10/11 (Extended Data Fig. 1b) as it shares a predicted catalytic aspartate and other active site sequences only conserved among ACAD10/11 homologs. ACAD10 also has a haloacid dehalogenase-like domain (Fig. 1b) that lacks homology to any protein in the *lva* operon and was not evaluated further here. Overall, our sequence conservation analysis supports ACAD10/11 as bifunctional homologs of LvaA and LvaC and candidates for the missing enzymes of mammalian 4-HA catabolism.

We next purified recombinant wild-type (WT) and two mutant versions of ACAD10/11 to test their in vitro activities against a 4-hydroxyacyl-CoA substrate (Extended Data Fig. 1c,d). In the first mutant (K*), we replaced the catalytic aspartate of the kinase domain with alanine, thereby eliminating kinase activity but leaving ACAD activity intact. In the second mutant (A*), we replaced the catalytic aspartate of the ACAD domain with asparagine, eliminating the ACAD activity but leaving kinase activity intact. We used recombinant LvaE, an acyl-CoA synthetase encoded in the *lva* operon[22], to produce 4-HV-CoA as a substrate for in vitro reactions. The activities of ACAD10/11 were evaluated by measuring intermediates expected to originate from 4-HV-CoA using liquid chromatography (LC)–MS and high-performance LC (HPLC) (Extended Data Fig. 1e). The WT version of each enzyme converted 4-HV-CoA into 2-pentenoyl-CoA as a final product while also producing differing levels of the 4-phosphovaleryl-CoA (4-PV-CoA) intermediate (Fig. 1c and Extended Data Fig. 1f,g). The K* mutants demonstrated little to no substrate consumption, whereas the A* mutants generated the 4-PV-CoA intermediate but were unable to complete conversion into 2-pentenoyl-CoA. We also observed formation of 3-HV-CoA in our reactions but determined it to be a nonenzymatic byproduct of 2-pentenoyl-CoA (Extended Data Fig. 2a).

Our results demonstrate that ACAD10/11 are sufficient to convert 4-HV-CoA into 2-pentenoyl-CoA, a conventional FAO intermediate, thereby nominating them as the missing enzymes necessary to execute the major 4-HA catabolic route described previously[18]. Our results further demonstrate that the ACAD10/11 kinase reaction precedes the ACAD reaction, indicating that the ACAD domains of these enzymes must accommodate a phosphorylated substrate. Together, this implies that the ACAD10/11 ACAD domains are functionally distinct from other ACAD enzymes, consistent with evolutionary analyses demonstrating that ACAD10/11 share a more recent common ancestor with each other than with other ACADs[24].

To test this hypothesis, we compared the activities of ACAD10/11 with three recombinantly purified ACAD enzymes involved in FAO, ACADS, ACADM and ACADVL, which have well-established specificity for short-chain, medium-chain and long-chain or very-long-chain acyl-CoA substrates, respectively. First, we tested the ability of these enzymes and ACAD10/11 to oxidize fatty acyl-CoAs ranging in length from 6 to 22 carbons into their 2-enoyl-CoA products using a ferrocenium assay[25]. ACADS, ACADM and ACADVL each exhibited clear activity consistent with their established specificity (Fig. 1d). However, ACAD10/11 showed no activity against any substrate under these conditions. Previous work suggested that ACAD11 can accommodate unsubstituted longer-chain substrates similar to those tested here[26]. However, these experiments were performed using bacterial lysates expressing a truncated form of ACAD11. While it is possible that ACAD10/11 may retain partial activity against classical FAO substrates under different conditions, the head-to-head comparison here demonstrates that this is unlikely.

We next compared the activities of each enzyme against 4-phosphohexanoyl-CoA (4-PH-CoA), an intermediate that can be generated by the kinase domains of ACAD10/11 from 4-hydroxyhexanoyl-CoA. We produced 4-PH-CoA using the ACAD11 A* mutant because it is unable to act upon the phosphorylated intermediate. WT ACAD10/11 provided 4-PH-CoA successfully converted it into the 2-hexenoyl-CoA product (Fig. 1e), indicating that the phospho-intermediate does not need to be directly channeled to the ACAD domain from the kinase domain. However, none of the other conventional ACADs showed any detectable activity against 4-PH-CoA. Together, these results demonstrate that the ACAD domains of ACAD10/11 are functionally distinct from other ACADs and likely do not contribute appreciably to the catabolism of conventional fatty acids.

To better understand the interplay between the kinase and ACAD domains, we evaluated the kinetics of their activities. To do so, we obtained pure 4-HV-CoA and 4-PV-CoA substrates (Methods) and used them to perform Michaelis–Menten kinetic analyses of ACAD11's kinase and ACAD domains, respectively (Fig. 1f,g and Extended Data Fig. 2b). The kinase domain showed both a higher $K_m$ (86.3 ± 26 μM) and a higher $V_{max}$ (19.4 ± 2.0 μM min$^{-1}$) than the ACAD domain ($K_m$ = 51.4 ± 30 μM; $V_{max}$ = 11.1 ± 1.6 μM min$^{-1}$). These results suggest that the kinase domain does not bind its substrate as readily as the ACAD domain binds the phospho-intermediate but that the ACAD domain-mediated reaction is likely rate limiting in the overall conversion of 4-hydroxyacyl-CoAs to 2-enoyl-CoAs.

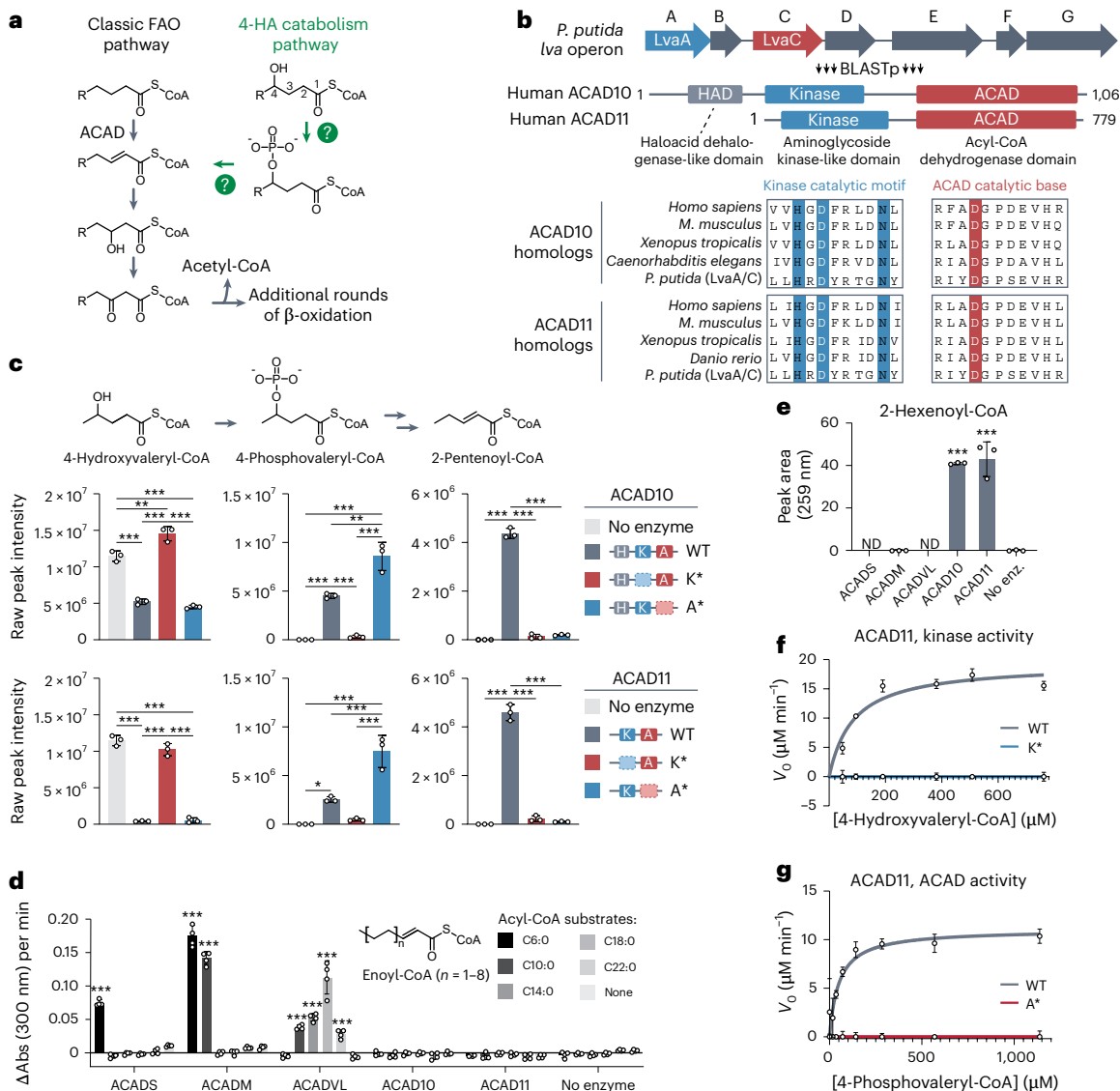

**Fig. 1 | ACAD10/11 process 4-HA substrates in vitro. a**, Model of the major 4-HA catabolism pathway and its integration into FAO as previously described[18,19]. Question marks indicate uncharacterized enzymatic steps in mammals. **b**, Multiple-sequence alignment demonstrating conservation between LvaA and LvaC of the *lva* operon[22] and eukaryotic ACAD10/11 homologs. **c**, LC–MS analysis of in vitro enzyme reactions with 4-HV-CoA as the starting substrate. Top: the raw signal intensities and chemical structures of 4-HV-CoA, 4-PV-CoA and 2-pentenoyl-CoA. Bar graphs represent data from reactions with recombinant ACAD10$^{N\Delta34}$ (top) and ACAD11 (bottom). The same no-enzyme control data are displayed on both sets of graphs for clearer comparison. Data are expressed as the mean ± s.d. (*n* = 3 technical replicates). **d**, Ferrocenium ion assay with recombinant ACAD family members and unsubstituted acyl-CoA substrates of various lengths. Top: general structure of the final product, enoyl-CoA. Initial velocity of the enzyme reaction corresponds to the change in absorbance

over time (mean ± s.d.; *n* = 4 technical replicates). **e**, UV–vis quantification of 2-hexenoyl-CoA produced from 4-PH-CoA substrate (mean ± s.d.; *n* = 3 technical replicates). **f,g**, Michaelis–Menten plots of ACAD11 kinase (**f**) and ACAD domain (**g**) activities against synthetic 4-HV-CoA and 4-PV-CoA, respectively. Data are normalized to K* and A* inactive mutant controls. Each data point represents the mean ± s.d. (*n* = 3 technical replicates; *n* = 6 technical replicates for WT in **f**; only averages are shown for clarity). Statistical analysis was conducted using a one-way analysis of variance (ANOVA) with Tukey's multiple-comparisons test (**c**), two-way ANOVA with Šídák's multiple-comparisons test comparing the activity of each enzyme against an acyl-CoA substrate to the respective no-enzyme control (**d**) or one-way ANOVA with Dunnett's multiple-comparisons test comparing all reactions with the no-enzyme control (**e**). \**P* < 0.05, \*\**P* < 0.01 and \*\*\**P* < 0.001. ND, not detected.

Lastly, we evaluated whether ACAD10/11 are necessary for the major 4-HV catabolic pathway by performing stable isotope tracing in Hepa1-6 cells. To do so, we cultured WT and CRISPR–Cas9-mediated *Acad10;Acad11* double-knockout (DKO) cells with [$^{13}C_5$]-4-HV and measured both total and labeled (M + 3) propionyl-CoA (Fig. 2a). M + 3 labeling of propionyl-CoA is expected only if [$^{13}C_5$]-4-HV is catabolized through the 4-PV-CoA intermediate[19] (Extended Data Fig. 2c). Each cell line possessed comparable levels of total propionyl-CoA (Fig. 2b); however, only WT Hepa1-6 cells produced M + 3 propionyl-CoA (Fig. 2c). Consistent with this, none of the distinctive 4-PV-CoA intermediate

was observed in the DKO cells even when cultured with high amounts of 4-HV (Fig. 2d). Overall, these results demonstrate that ACAD10/11 are sufficient for 4-HA conversion into physiological FAO substrates in vitro and are necessary for the major 4-HA catabolic pathway in Hepa1-6 cells.

## ACAD11 structure reveals determinants of molecular function

To further explore the bifunctionality of ACAD10/11 at the molecular level, we initiated cryo-electron microscopy (cryo-EM) experiments with the ACAD11 K* and A* mutant proteins purified from

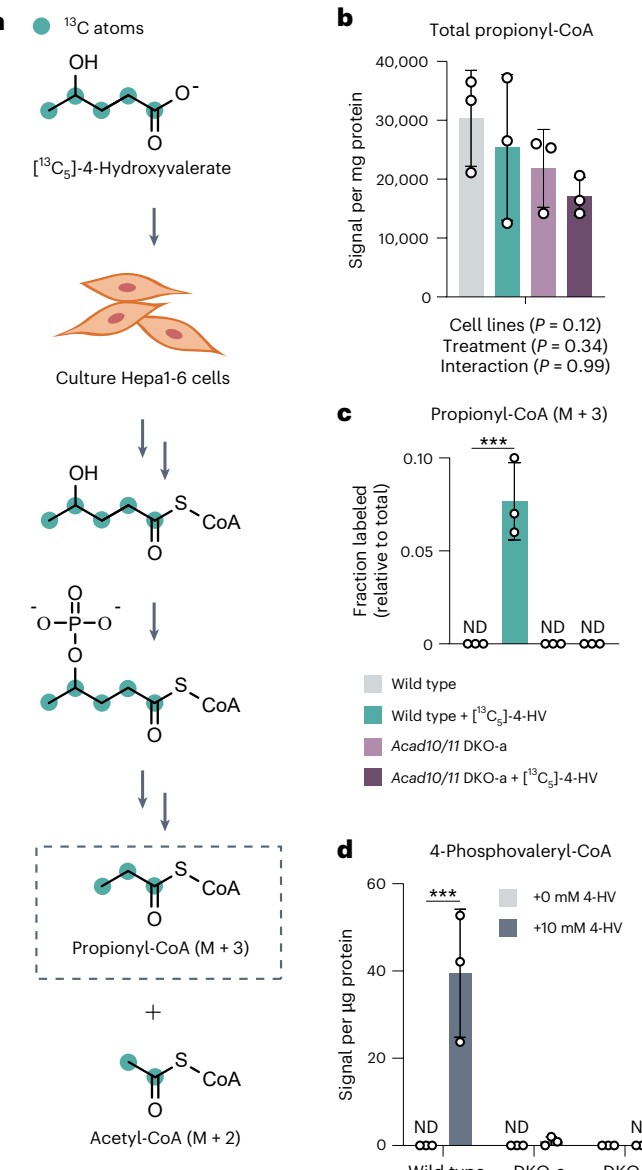

**Fig. 2 | ACAD10/11 process 4-HA substrates in cells. a**, Stable isotope tracing strategy to study [$^{13}C_5$]-4-HV catabolism specifically through the major pathway. **b**, Total abundance of all propionyl-CoA isotopologs. Data represent the averaged sum of raw signal intensities of each isotopolog. **c**, Fractional abundance of M + 3 propionyl-CoA relative to total propionyl-CoA pool in Hepa1-6 cells cultured with 5 mM [$^{13}C_5$]-4-HV. **d**, 4-PV-CoA abundance detected in Hepa1-6 cells treated with 10 mM unlabeled 4-HV. These same data are also shown in Fig. 4c. Metabolomic results in **b**–**d** are expressed as the mean ± s.d. ($n$ = 3 technical replicates). Statistical analysis was conducted using a two-way ANOVA with Šídák's multiple-comparisons test (**b**–**d**). *$P < 0.05$, **$P < 0.01$ and ***$P < 0.001$.

## Table 1 | Cryo-EM data collection, refinement and validation statistics

| | ACAD11 K* 4-HV-CoA (EMD-42954) (PDB 8V3U) | ACAD11 A* 4-PV-CoA (EMD-42955) (PDB 8V3V) |
|---|---|---|
| **Data collection and processing** | | |
| Magnification | 59k | 59k |
| Voltage (kV) | 300 | 300 |
| Electron exposure (e⁻ Å⁻²) | 54 | 54 |
| Defocus range (μm) | −1.0 to −2.4 | −1.0 to −2.4 |
| Pixel size (Å) | 1.1 | 1.1 |
| Symmetry imposed | $D_2$ | $D_2$ |
| Initial particle images (number) | 722,751 | 1,294,186 |
| Final particle images (number) | 242,249 | 297,619 |
| Map resolution (Å) | 2.6 | 3.6 |
| FSC threshold | 0.143 | 0.143 |
| Map resolution range (Å) | 2.2–3.0 | 2.5–6.5 |
| **Refinement** | | |
| Initial model used (PDB code) | AlphaFold and PDB 2WBI | This study and AlphaFold |
| Model resolution (Å) | 2.7 | 3.7 |
| FSC threshold | 0.5 | 0.5 |
| Map sharpening B factor (Å²) | −137.8 | −144.7 |
| Model composition | | |
| Nonhydrogen atoms | 11,928 | 14,468 |
| Protein residues | 1,576 | 2,264 |
| Ligands | 4 | 4 |
| B factors (Å²) | | |
| Protein | 29.19 | 60.32 |
| Ligand | 16.77 | 78.76 |
| R.m.s. deviations | | |
| Bond lengths (Å) | 0.003 | 0.002 |
| Bond angles (°) | 0.461 | 0.447 |
| **Validation** | | |
| MolProbity score | 1.15 | 1.36 |
| Clash score | 3.55 | 4.66 |
| Poor rotamers (%) | 0.71 | 0.47 |
| Ramachandran plot | | |
| Favored (%) | 98.97 | 97.36 |
| Allowed (%) | 1.03 | 2.64 |
| Disallowed (%) | 0 | 0 |

*Escherichia coli*. To assess whether substrate interactions affect protein structure, these mutants were incubated with 4-HV-CoA and 4-PV-CoA, respectively, before preparation. Negative-stain transmission EM analysis revealed homogeneous, monodispersed particles for each sample. The samples were vitrified, imaged and reconstructed using a three-dimensional single-particle cryo-EM analysis (Extended Data Figs. 3 and 4, Table 1 and Methods). Analysis of the ACAD11 K* mutant particles revealed an intact, tetrameric structure but only the ACAD domains were resolved (Extended Data Fig. 5a). The tetramer is consistent with resolved structures of other ACADs, including an unpublished structure of a truncated human ACAD11 that comprises

the ACAD domain only (Protein Data Bank (PDB) 2WBI). However, both domains were resolved in the ACAD11 A* mutant structure, which revealed ordered interactions between the kinase domains and the ACAD tetramer (Fig. 3a and Extended Data Fig. 5b).

The core ACAD11 kinase fold is similar to that of well-characterized PKLs[23,27,28] (Extended Data Fig. 5c), whereby the N lobe consists of an extended β-sheet and a single α-helix (αC) and the C lobe comprises a series of α-helices and β-strands. The kinase domains form dimers at the αC helix interface consistent with those seen in the related eukaryotic choline kinases[29]. To explore how the kinase domain might bind an acyl-CoA substrate, we modeled 4-hydroxyoctanoyl-CoA (4-HO-CoA) in the ACAD11 A* kinase domain's active site pocket (Methods). The atomistic molecular dynamics (MD) trajectories

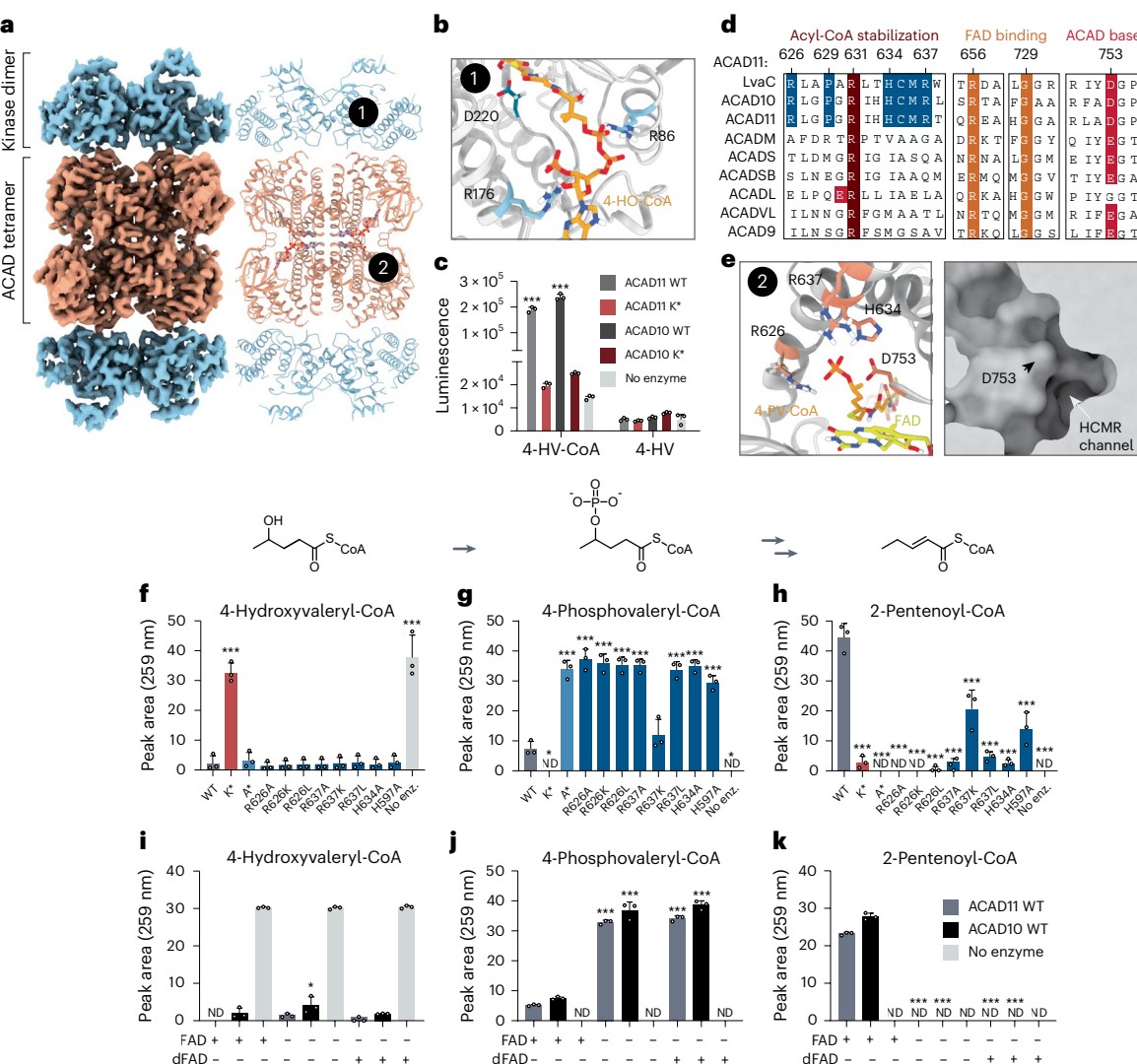

**Fig. 3 | Cryo-EM of ACAD11 reveals structural determinants of molecular function. a**, Cryo-EM map and model of the ACAD11 A* mutant tetramer incubated with 4-PV-CoA. Kinase and ACAD domains are colored blue and orange, respectively. One molecule of FAD (sticks) is present in each ACAD domain active site. **b**, MD modeling of 4-HO-CoA (orange sticks) in the kinase domain active site. Conserved residues implicated in substrate binding are presented as light-blue sticks. Catalytic D220 is shown as dark-blue sticks. **c**, Kinase activity of WT ACAD10/11 or K* mutants with 4-HV-CoA or nonacylated 4-HV substrates. **d**, Sequence alignments of *M. musculus* ACAD10/11 with *P. putida* LvaC and other *M. musculus* ACAD family members. Highly conserved residues involved in acyl-CoA stabilization, FAD binding and catalytic activity are colored brown, orange and red, respectively. Highly conserved ACAD domain residues possibly involved with 4-phosphoacyl-CoA substrate recognition are highlighted in blue. **e**, MD modeling of 4-PV-CoA (orange sticks) in the ACAD domain active site.

Key interacting residues (R626, R637, D753 and H634) are presented as sticks. Right, representation of the ACAD domain active site pocket surface. **f–h**, UV–vis absorbance (259 nm) of 4-HV-CoA (**f**), 4-PV-CoA (**g**) and 2-pentenoyl-CoA (**h**) in reactions containing WT ACAD11 and catalytically inactive mutants. **i–k**, UV–vis absorbance (259 nm) of 4-HV-CoA (**i**), 4-PV-CoA (**j**) and 2-pentenoyl-CoA (**k**) in reactions containing 5 μM FAD or 5-deazaFAD (dFAD). Data in **f–k** are presented as the mean ± s.d. (*n* = 3 technical replicates). Statistical analysis was conducted using a two-way ANOVA with Šídák's multiple-comparisons test comparing all conditions to the respective no-enzyme control (**c**), one-way ANOVA with Dunnett's multiple-comparisons test comparing all mutants to the WT control (**f–h**) or two-way ANOVA with Šídák's multiple-comparisons test comparing all conditions to the respective FAD-containing reaction controls (**i–k**). *$P < 0.05$, **$P < 0.01$ and ***$P < 0.001$.

revealed a stable hydrogen bond between the catalytic D220 and the substrate's 4-OH group, assisted by a hydrogen bond between N225 and the D220 carboxylate[30] (Extended Data Fig. 5d). We observed that a substantial portion of the CoA moiety associated with residues inside a cavity adjacent to the active site opening. These included R86, which neighbors the canonical glutamate of the αC helix, and R176, which resides in a long region located between the αE helix and the catalytic subdomain (Fig. 3b). This region often contributes to substrate recognition for members of the APH3 family[23]. Consistent with this, neither ACAD10 nor ACAD11 was able to phosphorylate the free acid form of 4-HV, indicating that the CoA moiety is necessary for substrate recognition (Fig. 3c). Together, these data demonstrate that the

ACAD10/11 kinase domain is poised to recognize 4-hydroxyacyl-CoA species to generate a phosphorylated substrate for further processing by the ACAD domain.

The active sites of the tetrameric ACAD domain show density that is consistent with bound flavin adenine dinucleotide (FAD). A cluster of amino acids neighboring this cofactor, which are uniquely conserved between ACAD10/11 and LvaC, may explain ACAD11's ability to accommodate a phosphorylated substrate (Fig. 3d). R626, H634 and R637 are found on a single α-helix within the ACAD domain active site that is directly adjacent to the catalytic D753 (Fig. 3e). These residues point toward D753 and the open cavity of the active site. To understand how these residues may interact with the 4-phosphoacyl-CoA

substrate, we modeled 4-phospho-octanoyl-CoA into the ACAD11 A\* ACAD domain (Methods). Atomistic MD simulations of the generated complex revealed a stable substrate binding position within the active site. Regions outside the active site interact with the CoA moiety of the substrate analogously to a substrate-bound medium-chain ACAD (PDB 1UDY)[31], including a small pocket that binds the adenosine group at the protein surface. Inside the active site, the α-carbons and β-carbons of the substrate are positioned between the catalytic D753 base and FAD, as seen in other ACADs. This orientation is facilitated by R631, which is conserved across ACAD family members (Fig. 3d). The 4-phospho group is directly stabilized by salt bridges with H634, R637 and R626—residues not found in conventional ACADs—while the hydrophobic tail itself extends toward a broad but closed cavity with a largely hydrophobic surface, suggesting the domain can accommodate much longer substrates (Fig. 3d,e). To investigate this region experimentally, we purified a series of ACAD10/11 constructs with substitutions of these conserved residues and measured their activities against 4-HV-CoA (Extended Data Fig. 5e and Supplementary Fig. 3). As predicted, disruptions to this region ablated the ACAD domain activity, resulting in phospho-intermediate accumulation (Fig. 3f–h). Nonconservative substitutions of R626, H634 and R637 to alanine or leucine were most disruptive to ACAD domain activity. Interestingly, the substitution of R637 to lysine only partially impaired turnover, suggesting that this site is permissive to a semiconservative change. Substitution of H597, another histidine proximal to FAD but not adjacent to the HCMR region, only partially reduced activity, suggesting that it is not critical for catalysis but may be generally important for active site structure.

In typical ACADs, FAD is a requisite cofactor that is reduced to FADH$_2$ concomitant with substrate oxidation[32]. However, the conversion of 4-PV-CoA to 2-pentenoyl-CoA does not involve a permanent change in oxidation state. To test whether ACAD10/11 require FAD for activity, we performed enzymatic assays on 4-HV-CoA with increasing concentrations of FAD, including submolar quantities, and quantified the substrate, intermediate and product using HPLC. We observed that both ACAD10/11 require FAD to convert the phospho-intermediate into the final 2-enoyl-CoA product and that activity diminished as the FAD concentrations decreased (Extended Data Fig. 5f).

To understand whether FAD is redox-active during this reaction, we substituted FAD with 5-deazaFAD, which limits the redox capacity of the cofactor while keeping binding and structural roles intact. ACAD10/11 showed no detectable ACAD domain activity in the presence of 5-deazaFAD, suggesting that these enzymes still require redox-active FAD to convert 4-phosphoacyl-CoA to 2-enoyl-CoA (Fig. 3i–k). These results suggest that ACAD10/11 use FAD in an atypical manner (Discussion). Overall, these structure–function analyses reveal multiple distinctive features of ACAD10/11 that enable their processing of 4-hydroxyacyl-CoA and 4-phosphoacyl-CoA substrates.

## ACAD10/11 catabolize distinct 4-HAs in different organelles

Our enzyme assays indicated that ACAD10/11 had comparable activities against the substrates tested in vitro; however, it is unclear whether they act redundantly in cellular 4-HA catabolism. Other acyl-CoA-processing enzymes with similar inherent substrate specificities are known to reside in distinct subcellular locations where they encounter different substrates[33]. ACAD10 has a predicted N-terminal mitochondrial targeting sequence[34] (MTS) and ACAD11 has a predicted C-terminal type 1 peroxisomal targeting signal (PTS)[35] (Fig. 4a); however, previous localization studies have provided conflicting results[26,36,37].

To clarify their subcellular localizations, we ectopically expressed human ACAD10–FLAG and FLAG–ACAD11 in COS7 and U2OS cells and imaged their distribution. Our imaging and fluorescence emission line quantification showed distinct localization of ACAD10/11 to separate organelles in both cell lines; ACAD10–FLAG localized to mitochondria and FLAG–ACAD11 localized to peroxisomes (Fig. 4a and Extended

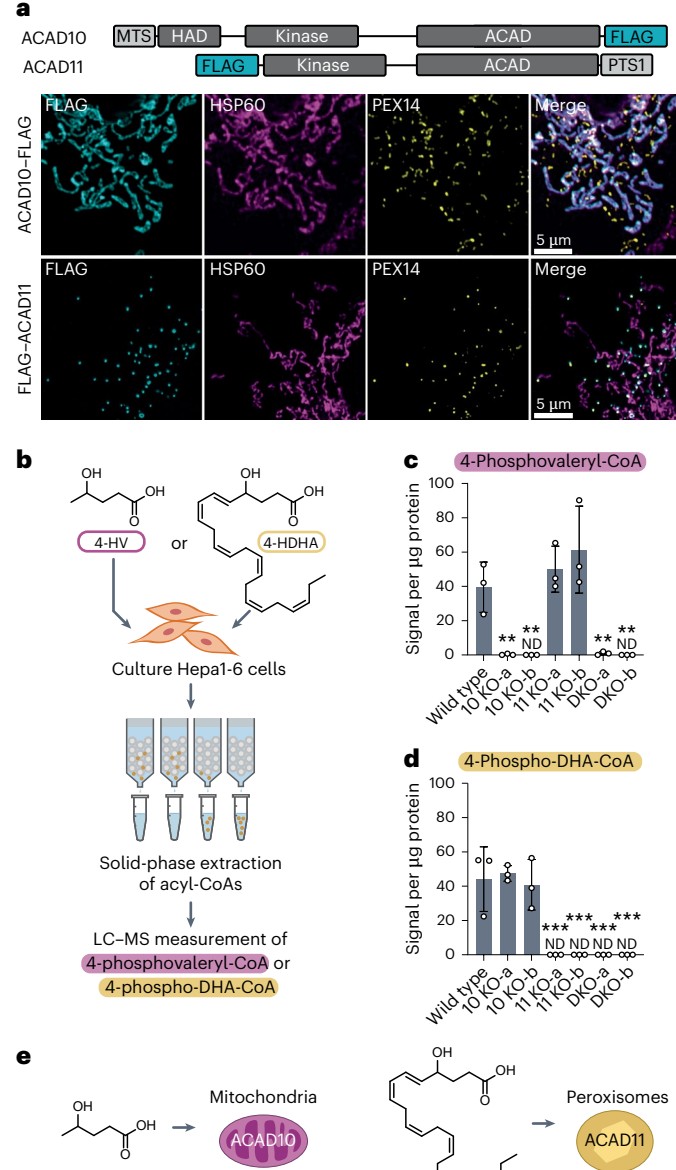

**Fig. 4 | ACAD10/11 mediate catabolism of distinct 4-HAs in different organelles. a**, Immunofluorescence imaging of overexpressed human ACAD10–FLAG and FLAG–ACAD11 in COS7 cells. Localization of FLAG-tagged constructs (cyan) is indicated by overlap in fluorescence intensity with mitochondrial (HSP60; magenta) and peroxisomal (PEX14; yellow) markers. Top, natural positioning of MTS and PTS on the full-length isoforms of ACAD10/11. **b**, Experimental workflow to evaluate 4-HA metabolism in Hepa1-6 cells. **c**, 4-PV-CoA levels detected in Hepa1-6 cells treated with 0 mM or 10 mM unlabeled 4-HV. **d**, 4-Phospho-DHA-CoA levels detected in Hepa1-6 cells treated with unlabeled 100 μM 4-HDHA conjugated to 1% fatty-acid-free BSA (w/v). Results in **c,d** are expressed as the mean signal intensity (normalized to total protein content) ± s.d. (*n* = 3 independent experiments). **e**, Model of organellar 4-HA catabolism. Statistical analysis was conducted using a one-way ANOVA with Dunnett's multiple-comparisons test comparing all cell lines to the WT control (**c,d**). \**P* < 0.05, \*\**P* < 0.01 and \*\*\**P* < 0.001.

Data Fig. 6a–c). Localization of these paralogs to the mitochondria and peroxisomes was disrupted when the positions of their FLAG tags were swapped to the opposite termini (Extended Data Fig. 6d).

In metazoans, FAO is conducted either in mitochondria or in peroxisomes depending on the length of the fatty acid[38]. Thus, we hypothesized that ACAD10/11 are tasked with catabolizing 4-HA species of different lengths according to their organellar localization.

Canonically, mitochondria catabolize short-chain to long-chain fatty acids (approximately 4–18 carbons) and peroxisomes catabolize very-long-chain fatty acids (>20 carbons). Using Hepa1-6 CRISPR–Cas9 KO cell models lacking functional ACAD10 and/or ACAD11, we evaluated the necessity of either ACAD10 or ACAD11 in the catabolism of two different 4-HAs, 4-HV (5 carbons long) and 4-HDHA (22 carbons long), by monitoring the formation of their respective 4-phosphoacyl-CoA intermediates (Fig. 4b). All cell lines supplemented with the 4-HAs produced the corresponding 4-hydroxyacyl-CoA intermediates (Extended Data Fig. 6e,f). When given 4-HV, cells specifically lacking functional ACAD10 were unable to generate 4-PV-CoA, whereas cells lacking ACAD11 produced 4-PV-CoA at levels comparable to WT (Fig. 4c). Reciprocally, when supplemented with 4-HDHA, cells lacking ACAD11 failed to produce detectable 4-phospho-DHA-CoA, whereas cells lacking functional ACAD10 performed like the WT control (Fig. 4d). These results suggest that ACAD10 functions in mitochondrial short-chain 4-HA catabolism and ACAD11 functions in peroxisomal very-long-chain 4-HA catabolism (Fig. 4e).

### Acad11-KO mice have aberrant 4-HA homeostasis and fat gain

Our collective enzymology and cellular work establish ACAD10/11 as enzymes capable of 4-HA catabolism but whether they control 4-HA homeostasis in vivo is unknown. In addition, the physiological importance of 4-HA catabolism remains cryptic. To address these questions, we established a whole-body KO of *Acad11* in the C57BL/6 mouse background (Extended Data Fig. 7a,b). All genotypes were born near the expected Mendelian frequency (Extended Data Fig. 7c) and had similar survival throughout our experiments.

To date, no human diseases have been causally linked to mutations in *ACAD10* or *ACAD11*, although multiple genome-wide association studies (GWAS) have suggested their associations with metabolic and cardiovascular functions[39–44]. Additionally, the human exome–trait association database Genebass[45] reports significant sex-specific fat mass traits associated with rare ACAD11-coding variants (Fig. 5a). Given these observations, we assessed various molecular and physiological phenotypes in WT and *Acad11*-KO mice fed either standard chow or a high-fat diet (HFD).

FAO enzymatic deficiencies often result in the systemic accumulation of unprocessed FAO substrates and intermediates[46]. As such, we began by developing a targeted LC–MS method to measure various hydroxylated free fatty acids in mouse plasma (Supplementary Table 4). Consistent with our in vitro biochemical findings, various 4-HA species, such as 4-OH C10 and 4-OH C12, were significantly elevated in the plasma of male and female *Acad11*-KO mice fed either diet (Fig. 5b,c and Extended Data Fig. 7d,e). Notably, we found that 4-HAs in particular were perturbed in *Acad11*-KO mice while fatty acids with hydroxylation at other carbons were unchanged. For example, 4-OH C10 and 4-OH C12 were among the most consistently elevated 4-HAs in KO mice under both standard chow and HFD feeding, whereas the corresponding 3-OH and 5-OH isomers were unperturbed across all genotypes in both sexes (Extended Data Fig. 7f–i).

During the 12-week HFD regimen, we observed that *Acad11*-KO females gained more weight than their littermate controls (Fig. 5d,e). Strikingly, these mice also had increased white adipose tissue (WAT) depots (Fig. 5f,g), indicating that body weight gain was driven by fat accumulation, consistent with the human female-specific fat mass associations (Fig. 5a). The total body and individual tissue weights were unchanged for male mice on HFD and for both sexes fed standard chow (Extended Data Fig. 8a–k). In addition, we observed that HFD-fed *Acad11*-KO females had smaller kidneys and hearts but no change in liver size (Fig. 5h–j). We assayed various markers of liver and kidney pathology and function in the plasma after HFD feeding (for example, alanine transaminase, aspartate transaminase and blood urea nitrogen) but saw no significant differences in *Acad11*-KO mice (Extended Data Fig. 8l–q).

To follow up these observations and begin exploring the role of ACAD11 in adipocytes, we isolated preadipocytes from WT and *Acad11*-KO female mice and assessed their differentiation ex vivo. Following differentiation, *Acad11*-KO adipocytes exhibited lower lipid droplet abundance (Fig. 5k,l). However, the lipid droplets in the *Acad11*-KO adipocytes were larger in size (Fig. 5m,n), consistent with growth through hypertrophy rather than growth through hyperplasia. The *Acad11*-KO adipocytes also had reduced expression of select transcriptional markers of differentiation, such as *Glut4* and *Tle3*, a transcriptional coregulator of adipogenesis[47] (Fig. 5o,p and Extended Data Fig. 8r–t). In addition, these KO adipocytes had lower mRNA expression of the adipokines adiponectin and leptin (Fig. 5q,r), which are established regulators of insulin responsiveness, inflammation, satiety and energy expenditure[48,49].

On the basis of our animal model results, we propose that ACAD11 functions to eliminate 4-HAs in vivo and that dysfunctional ACAD11 activity leads to aberrant HFD-induced WAT accumulation in a sexually dimorphic manner. However, we acknowledge that our study is limited as we only measured the free acid forms of 4-HAs in mouse plasma and it remains unclear how 4-HAs regulate physiology and tissue-specific functions. Nonetheless, our overall study nominates ACAD10/11 as the primary gatekeepers of mammalian 4-HA metabolism, thereby providing a foundation to further investigate the mechanisms by which they catabolize 4-HAs, identify new 4-HAs connected to their activities and explore the pathophysiology that may result from their disruption.

## Discussion

Here, we demonstrate that ACAD10/11 catabolize 4-HAs through the major pathway described in literature[18,19]. Notably, while this paper was under revision, an independent study reported biochemical findings highly consistent with our conclusions here and in our original archived paper[50,51]. ACAD10/11 are evolutionarily divergent members of the ACAD family[24], possessing both an N-terminal kinase domain and unconventional hydrophilic residues in the ACAD active site. These features allow them to phosphorylate and process 4-hydroxylated acyl-CoA substrates, ultimately removing the hydroxy group and producing 2-enoyl-CoAs that can enter the FAO cycle. The presence of hydrophilic residues is uncommon in ACAD active sites, which are typically hydrophobic to accommodate neutral-charged, long-chain acyl tails[32]. However, our structure-guided analysis of ACAD11 revealed that the ACAD10/11-specific HCMR motif is essential for activity and exists proximal to FAD and the catalytic aspartate. We also identified a highly conserved arginine upstream of the HCMR motif (R626 for mouse ACAD11) that is likely required for stabilizing the negatively charged phosphate during its release. Interestingly, the decarboxylation activity of glutaryl-CoA dehydrogenase—another atypical member of the ACAD family—relies on an analogous arginine, which is thought to stabilize the negatively charged carboxylate group[52].

The precise catalytic mechanisms of both ACAD10/11 domains remain unclear. In ACAD11's ACAD domain, FAD is bound in the canonical position seen in other ACADs and is required for catalytic turnover of 4-phosphoacyl-CoA. Although 4-phosphorylated substrates and their corresponding 2-enoyl-CoA products have the same oxidation state, our experiments with 5-deazaFAD suggested that FAD still acts in a redox-active manner. However, FAD is not terminally reduced to $FADH_2$, as it does not require an electron acceptor to be regenerated after catalysis in vitro. We suggest a mechanism by which FAD is first reduced but later donates a hydride as part of the phosphoelimination mechanism to return to an oxidized state. Similar 'rebound' functionality for flavin cofactors has been observed in other enzymatic mechanisms, including those of chorismate synthase and a linoleic acid isomerase from *Propionibacterium acnes*[53,54]. Further work is required to clarify the mechanistic roles of FAD and the other unique active site features of ACAD10/11 revealed through our structure.

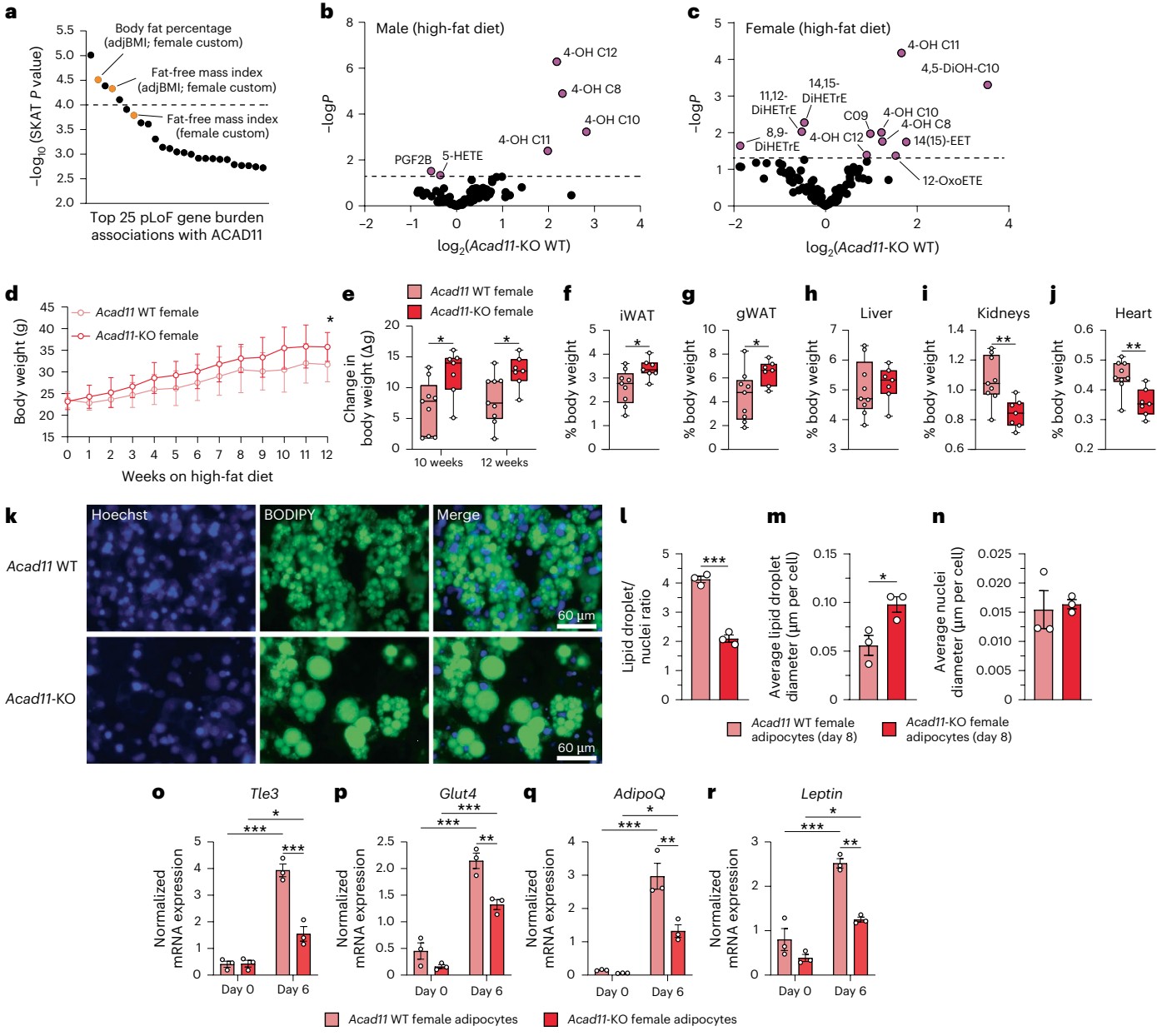

**Fig. 5 | *Acad11*-KO mice have aberrant 4-HA homeostasis and diet-induced fain gain. a**, Top 25 traits associated with predicted loss-of-function (pLoF) rare coding variants of human *ACAD11* (Genebass exome database). Traits are ranked by statistical significance as calculated by SKAT gene burden test ($-\log P \geq 4$ denotes a strong association). Orange circles are traits related to fat. adjBMI, adjusted to body mass index. **b,c**, Volcano plots depicting $\log_2$-transformed fold changes of 4-HAs and other hydroxylated lipids in HFD-fed *Acad11*-KO mice versus statistical significance. Male (**b**) and female (**c**) *Acad11*-KO mice were compared to littermate controls. Male WT, $n = 6$; male KO, $n = 5$; female WT, $n = 10$; female KO, $n = 7$. Mice were 3 months old at the start of HFD feeding. **d**, Body weight of female WT and *Acad11*-KO mice throughout 12 weeks of HFD feeding. Data are presented as the mean ± s.d. **e**, Changes in body weight after HFD feeding. **f–j**, Percentage body weight of iWAT (**f**), gWAT (**g**), liver (**h**), kidneys (**i**) and heart (**j**) of HFD-fed mice. In **d–j**, female WT, $n = 9$; female KO, $n = 7$. Data in **e–j** are shown as box plots. Center line, median; box limits, 25th to 75th percentiles; whiskers, minimum and maximum points. **k**, Immunofluorescence images of WT and *Acad11*-KO primary adipocytes 8 days after differentiation ex

vivo (representative of $n = 3$ independent biological replicates). Hoechst (blue), nuclei; BODIPY 493/503 (green), lipid droplets. Magnification, ×100; scale bar, 60 μm. **l–n**, Quantification of lipid droplet quantity (**l**), lipid droplet size (**m**) and nucleus size (**n**) of WT and *Acad11*-KO differentiated primary adipocytes. Data are expressed as the mean ± s.e.m. ($n = 3$ independent biological replicates; each replicate is the average of three technical replicate images, except for two WT samples, each of which is the average of two technical replicate images). The average lipid droplet diameter (**m**) and average nuclei diameter (**n**) are normalized to cell count. **o–r**, mRNA expression of *Tle3* (**o**), *Glut4* (**p**), *AdipoQ* (**q**) and *Leptin* (**r**) in WT and *Acad11*-KO primary adipocytes during differentiation. Data are normalized to *Rps3*. Bars represent the mean ± s.e.m. ($n = 3$ independent biological replicates). Statistical analysis was conducted using multiple two-sided $t$-tests (**b,c**), a two-way repeated-measures ANOVA with Šídák's multiple-comparisons test comparing *Acad11*-KO mice to WT mice at each time point (**d**), a two-sided Student's $t$-test (**e–j,l–n**) or a two-way ANOVA with Šídák's multiple-comparisons test (**o–r**). *$P < 0.05$, **$P < 0.01$ and ***$P < 0.001$.

ACAD10/11 are localized to mitochondria and peroxisomes, respectively. Although catalytically similar, our work in cultured cells and in vivo suggests that ACAD10/11 do not fully compensate for one another in the catabolism of 4-HAs. We propose that this subcellular partitioning broadens the range of 4-HA species that the cell can catabolize. 4-HAs detected thus far in human and rodent models

range from 4 to 22 carbons[12,14,15,17]; conventional catabolism of fatty acids spanning this range requires enzymes from both mitochondria and peroxisomes[38]. While our results argue that short-chain and very-long-chain 4-HA catabolism is confined to the mitochondria and peroxisomes, respectively, we observed that medium-chain and long-chain 4-HAs were consistently elevated in *Acad11*-KO mice under different diet conditions. These data suggest that the minimal length of 4-HAs selected and catabolized by the peroxisome is much shorter than that of other typical fatty acids (>20 carbons long). In addition, ACAD10/11 may function with colocalized FAO machinery to catabolize precursor fatty acids that are eventually shortened into 4-HAs, such as those with hydroxyl groups on even-numbered carbons[21]. For example, microorganisms that are fed ricinoleic acid (that is, 12-hydroxyoleic acid) produce 4-OH C10 as a catabolic intermediate that is further degraded[16].

However, why do cells need to catabolize 4-HAs? We speculate that ACAD10/11-dependent catabolism of 4-HAs may be relevant for the efficient clearance of oxygenated lipids, such as those produced during excessive oxidative stress or inflammation. The origins of 4-HAs are largely unknown; however, a few species may be produced from reactive aldehydes, such as 4-hydroxynonenal (4-HNE), that form during lipid peroxidation[55]. Previous work with rat liver tissue identified a nine-carbon 4-phosphoacyl-CoA intermediate derived from 4-OH C9, a downstream intermediate of 4-HNE[18,21]; thus, FAO has been proposed as a pathway for eliminating 4-HNE and protecting against its cytotoxic effects. In mammalian cells, mitochondrial and peroxisomal FAO coordinate the catabolism of oxylipins[56], which are signaling molecules that control inflammation and vascular response[57]. These include 4-HDHA[58] and others (for example, 12-HETE[59]), whose catabolism may yield 4-HA intermediate bottlenecks that are managed by ACAD10/11.

To date, no monogenic disorders stemming from ACAD10/11 have been identified. However, exome-based association analyses revealed a likely connection between *ACAD11* variants and female-specific fat metabolism[45]. Remarkably, this connection was borne out in our *Acad11*-KO mouse experiments, where females gained weight and exhibited increased inguinal WAT (iWAT) and gonadal WAT (gWAT) depots when fed an HFD. While the underlying mechanism of this phenotype requires further investigation, isolated adipocytes from the iWAT had larger lipid droplets along with diminished expression of adipocyte differentiation markers (for example, *Tle3*) and adipokines (that is, adiponectin and leptin), suggesting altered maturation of fat cells and hormonal action. Several GWAS have also identified single-nucleotide polymorphisms in both paralogs associated with coronary heart disease, obesity, diabetes and insulin resistance in diverse human populations[41–44]. In addition, previous work suggests links between *Acad10* and *Acad11* expression and insulin sensitivity in mice. A large-scale mouse GWAS revealed that adipose *Acad11* expression strongly correlates with the homeostatic model assessment of insulin resistance[39]. A separate study found that *Acad10*-KO mice (SvEv129/BL6 mixed background) exhibited higher fasting insulin levels and poor glucose tolerance[60]; however, a recent study demonstrated that these phenotypes were absent in a different *Acad10*-KO mouse model (C57BL/6J background)[61]. Collectively, these studies suggest many potential links between 4-HAs and human health. Our identification of ACAD10/11 as the primary gatekeepers of 4-HA metabolism will empower further studies to explore these connections.

## Online content

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

## Methods

### Animal models

Cryopreserved *Acad11*-KO (C57BL/6NJ) mouse sperm was purchased from The Jackson Laboratory (51098-JAX). In vitro fertilization of C57BL/B6J donor eggs with sperm was performed by the University of Wisconsin–Madison Animal Models Core. *Acad11* heterozygous progeny were inbred to maintain colony. WT littermate controls were included in all experiments. Breeder mice were maintained on a high-energy diet (Formulab Diet 5015). Experimental mice were maintained on standard chow diet (Formulab Diet 5008) unless specified otherwise. Mice were housed under a standard 12-h light–dark cycle, ambient temperature and humidity, and had free access to water and food. Our animal protocol was approved by the Institutional Animal Care and Use Committee of the College of Agricultural and Life Science at the University of Wisconsin–Madison.

### Mouse genotyping

Mouse tail clippings were digested (24–48 h, 55 °C) with 10 μl of 10 mg ml$^{-1}$ proteinase K in 400 μl of 20 mM Tris-HCl, 25 mM EDTA, 100 mM NaCl and 0.5% SDS (v/v), pH 8.0. The next day, samples were centrifuged to pellet undigested matter (16,300$g$, 5 min). Then, 390 μl of supernatant was mixed with 800 μl of cold isopropanol by inverting tubes 20 times. Samples were centrifuged to pellet precipitated genomic DNA (16,300$g$, 10 min, 4 °C). After aspirating the supernatants, precipitates were washed with 1 ml of 100% ethanol followed by 1 ml of 70% ethanol. After drying in a fume hood, genomic DNA precipitates were resuspended in 1× TE buffer, pH 8.0. *Acad11* genotypes were assessed using a three-primer PCR strategy with the 5× Multiplex PCR mix (New England Biolabs) to allow for the simultaneous detection of both WT and mutant alleles. PCRs were amplified by the following steps (steps 2–4 were repeated 35 times): (1) initial denaturation (1 min, 95 °C); (2) denaturation (30 s, 95 °C); (3) annealing (30 s, 53 °C); (4) elongation (1.5 min, 68 °C); and (5) final extension (5 min, 68 °C). PCRs were resolved on a 2% agarose gel stained with ethidium bromide before imaging.

### Cell culture

Hepa1-6 cells (mouse hepatoma; American Type Culture Collection, CRL-1830) were cultured in high-glucose DMEM containing 10% FBS (Gibco) and maintained at 37 °C and 5% $CO_2$. Cells used for experiments underwent no more than six passages. Monoclonal CRISPR KO Hepa1-6 cell lines were generated by the Genome Engineering and Stem Cell Center at Washington University in St. Louis using two different single-stranded guide RNAs that targeted early exons of *Acad10* and *Acad11*. After the selection of cells with indels predicted to introduce early stop codons and verification by next-generation sequencing, two monoclonal lines per gene KO combination were chosen. DKO cells were generated from *Acad11*-KO. Cell lines were verified to be free of *Mycoplasma* with the LookOut *Mycoplasma* qPCR detection kit (Sigma-Aldrich).

### Multiple-sequence alignment and homology searches

Computational search for *lva* operon enzyme homologs in mammals was conducted by querying their primary amino acid sequences against reference proteomes available on the UniProtKB/SwissProt databases (taxonomy restricted to mammalia) using the BLASTp 2.12.0+ algorithm. Subsequent analyses of LvaA and LvaC sequence conservation and coverage were performed with primary sequence sets containing National Center for Biotechnology Information HomoloGene sequences of ACAD10/11 or the full-length sequences of human acyl-CoA dehydrogenases. Multiple-sequence alignments of FASTA-formatted sequences were generated using the default settings of the MAFFT algorithm (version 7.505)[62]. The MAFFT alignment results were visualized with Jalview (version 2.11.2.5). The Pfam database was referenced to illustrate the human enzyme domain architecture in Fig. 1b.

### Plasmid cloning

For recombinant expression vectors, the coding sequences of mouse *Acad10* and *Acad11* were amplified from WT female C57BL/6J mouse liver tissue complementary DNA (cDNA) by PCR. Using Gibson assembly, *Acad10$^{NΔ34}$* (gene product lacking 34-aa N-terminal mitochondrial targeting sequence) and *Acad11* were cloned into different expression vectors. *Pichia pastoris* codon-optimized *Acad10$^{NΔ34}$* was ordered as gene fragments from GenScript and incorporated into the pPICZ-B vector with a 3′ GFP tag coding sequence and *Acad11* was cloned into the pE-SUMOstar vector with a 5′ SUMO tag coding sequence. pE-SUMO vectors for recombinant mouse ACADM$^{NΔ23}$, ACADS$^{NΔ29}$ and ACADVL$^{NΔ41}$ expression and purification were cloned similarly from templates purchased from GenScript. For ectopic gene expression in cell culture, human *ACAD10* and *ACAD11* sequences were cloned into pcDNA3.1 vectors with N-terminal or C-terminal FLAG sequences using similar strategies. Genes were amplified by PCR and inserted into the pcDNA3.1 backbone using Gibson assembly following digestion with EcoRI and HindIII. For all cloning work, 2× Q5 master mix (New England Biolabs) was used for high-fidelity PCR and 2× HiFi Gibson assembly mix (New England Biolabs) was used for amplicon assembly. Point mutations were generated using the Q5 site-directed mutagenesis kit (New England Biolabs). Primer sequences and plasmid construct information are listed in Supplementary Tables 1–3.

### Bacterial recombinant protein purification

For purification of *Mus musculus* ACAD11, ACADM$^{NΔ23}$, ACADS$^{NΔ29}$ and ACADVL$^{NΔ41}$, pE-SUMO constructs were transformed into B21-CodonPlus (DE3)RIPL competent *E. coli*. Colonies were picked and cultured in Luria–Bertani medium containing 50 μg ml$^{-1}$ kanamycin and 25 μg ml$^{-1}$ chloramphenicol and expression was induced with 100 μM IPTG (overnight with shaking, 18 °C). Cultures were harvested by centrifugation (4,000$g$, 20 min, 4 °C) and pellets were resuspended in lysis buffer (50 mM HEPES, 300 mM NaCl, 5% glycerol, 1 mM $MgCl_2$, 500 μM phenylmethylsulfonyl fluoride and 1 mM β-mercaptoethanol (β-ME), pH 7.4). For purification of ACAD11 constructs and ACADVL$^{NΔ41}$, 0.5% NP-40 IGEPAL (Sigma-Aldrich) and 0.5% Tween-20 (Fisher Scientific) were included in the lysis buffers, respectively. Cell pellets were suspended by hand into a slurry and sonicated on ice with a Bronson sonifier (75% amplitude, 20 s on and 5-min rest, three cycles). Lysates were clarified by a hard spin (30,000$g$, 30 min, 4 °C). The soluble fractions were incubated with 3–10 ml of Ni-NTA resin in buffer without detergent for 1 h with gentle tube rotation at 4 °C. Supernatants were removed and the resins were washed with buffer containing increasing concentrations of imidazole (10–50 mM), followed by elution with 25 ml of buffer containing 300 mM imidazole. The eluates were dialyzed overnight in the presence of Ulp1 protease (1 U per mg protein). The next day, the dialyzed protein samples were incubated with 1 ml of Ni-NTA resin (1 h, 4 °C) to remove excess free protease, SUMO tag and contaminants. The supernatants were collected and concentrated to 1 ml using an Amicon centrifugal filter with a 50-kDa cutoff. Concentrated proteins were separated by size-exclusion chromatography and fractions of the first major elution peak were assayed for the presence of recombinant protein using SDS–PAGE according to standard procedures. The purest fractions were pooled and exchanged into storage buffer (50 mM HEPES, 300 mM NaCl, 20% glycerol, 1 mM $MgCl_2$ and 1 mM β-ME, pH 7.4) before flash-freezing with liquid nitrogen and storing at −80 °C. The expression and batch purification of recombinant His–LvaE was performed as previously described except with the buffers used to purify recombinant ACAD11 (ref. [22]). SDS–PAGE gels were imaged using LI-COR Image Studio (version 5.2.5) and protein concentration was quantified using the Bradford assay (Bio-Rad).

### Yeast recombinant protein purification

For purification of *M. musculus* ACAD10$^{NΔ34}$, pPICZ-B-*Acad10$^{NΔ34}$*-GFP was transformed and expressed in *P. pastoris*. Individual clones were

selected and expression of the GFP-tagged protein was verified by western blot using a rabbit anti-GFP primary antibody (1:1,000; Abcam, ab6556) and IRDye 680RD goat anti-rabbit IgG secondary antibody (1:5,000; LI-COR Bio, 926-68071). Colonies with the highest expression for a given construct were used to inoculate 300 ml of YEPD medium containing 200 µg ml⁻¹ Zeocin and grown overnight with shaking (220 rpm, 30 °C). The next day, the full culture volume was used to inoculate 12 L of YEPD containing 200 µg ml⁻¹ Zeocin and grown for 24 h with shaking (220 rpm, 30 °C). Turbid cultures were centrifuged (20 min, 4,000$g$, room temperature) and the pellet was gently resuspended in buffer minimal methanol medium (200 mM potassium phosphate, 1× yeast nitrogen base, 40 µg biotin, 0.7% methanol and 100 µg ml⁻¹ zeocin, pH 6.0) to induce ACAD10$^{N\Delta34}$–GFP–His protein expression (220 rpm, 24 h, 24 °C). Cultures were harvested by centrifugation (4,000$g$, 20 min, 4 °C) and extruded from a wide-bore syringe into liquid nitrogen to create frozen droplets. Droplets were subjected to lysis using a CryoMill under frozen conditions. CryoMill lysate powder was gently resuspended in chilled lysis buffer and clarified by a hard spin (30,000$g$, 30 min, 4 °C). The supernatant was collected and incubated with 10 ml of Ni-NTA resin in standard purification buffer with gentle tube rotation (1 h, 4 °C). After removing supernatant, the resin was washed with increasing concentrations of imidazole (10–50 mM) followed by elution with buffer containing 300 mM imidazole. The eluate was dialyzed overnight with PreScission protease added (1 U per mg protein). The following day, the dialyzed protein was carefully stirred and diluted 20-fold in standard protein buffer without NaCl, resulting in a final NaCl concentration of 15 mM. The diluted mixture was concentrated to 25 ml using an Amicon centrifugal filter and run continuously (0.5 ml min⁻¹) over a 5-ml HiTrap SP fast flow fast protein LC column (Millipore Sigma) for 2–3 h to maximize protein capture. The protein was eluted from the column using a gradient of low-salt to high-salt buffer and SDS–PAGE was used to identify fractions containing the protein of interest, which were then pooled. The pooled fractions were stored overnight at 4 °C on ice before being concentrated to 1 ml using an Amicon centrifugal filter with a 50-kDa cutoff. Like the purification of recombinant proteins from bacteria, the concentrated protein was separated by size-exclusion chromatography. After verifying the presence of protein by SDS–PAGE (imaged using LI-COR Image Studio version 5.2.5), fractions of the main elution peak were pooled and exchanged into storage buffer before flash-freezing and storage at −80 °C.

### Production of acyl-CoA enzyme substrates

The 4-HAs used in this study were generated from γ-lactone precursors (Supplementary Table 6), unless otherwise specified. Lactones were saponified in 20–30% molar excess NaOH at 60–75 °C with intermittent vortexing. For in vitro experiments, short-chain acyl-CoAs were generated in situ by incubating 0.5 µM recombinant LvaE with equimolar amounts of ATP, CoA and free fatty acid (that is, valerate, 4-HV, 2-pentenoate or 3-pentenoate) for 30 min at room temperature (50-µl volume). Reactions containing acyl-CoA products were used directly as substrates for in vitro activity assays and standards for HPLC method development.

### In vitro enzyme activity assays

For LC–MS analysis, 100-µl reactions were prepared with 50 µM ATP, 50 µM FAD, 50 µM acyl-CoA substrate and 0.5 µM recombinant protein. Reactions were then rotated at 37 °C for 30 min. Reactions were quenched with 100 µl of cold quenching solvent (methanol–water (1:1, v/v) containing 5% acetic acid). Samples were centrifuged to pellet any precipitate (16,300$g$, 10 min, 4 °C). Each supernatant was transferred to a new 1.5-ml tube and dried in a speed vacuum concentrator overnight. Samples were stored at −80 °C before reconstitution in 500 µl LC–MS-grade water before analysis. For HPLC analysis, a similar protocol was followed with minor changes. Reactions were assembled

with 500 µM ATP, 5 µM FAD, 50 µM acyl-CoA substrate and 0.5 µM recombinant protein unless otherwise noted. The final assay volume was 25 µl and assays were quenched with 25 µl of quenching solvent. Quenched reaction mixtures were transferred directly to amber glass vials with inserts. HPLC elution standards were purchased or synthesized with enzymes and diluted in quenching solvent. All conditions were measured in triplicate and tested alongside negative control reactions lacking recombinant enzyme.

### HPLC analysis of in vitro assay mixtures

HPLC analysis of acyl-CoA species was performed similarly to previously published methods[63]. Before sample runs, the column (Betasil C18, 100 × 2.1 mm, 3-µm particle size; Thermo Scientific) was washed with 100 ml of solvent B (methanol) followed by 100 ml of solvent A (20 mM ammonium acetate and methanol (94:6, v/v), pH 7.0). A methanol blank was injected at the start of each running period. The injection volume for all samples, blanks and standards was 15 µl. The flow rate for the method was 0.5 ml min⁻¹ and the total duration of each individual run was 1 h. The solvent percentages were as follows for solvent A: 10.5 min of a gradient from 100% to 94%, 7.5 min at 94% isocratic, 27 min of a gradient from 94% to 25%, 2 min of a gradient from 25% to 100% and 13 min of 100% isocratic. Acyl-CoA peaks were quantified by postcolumn ultraviolet–visible light (UV–vis) absorbance measurements at 259 nm. Peak identities were verified using corresponding LvaE-generated standards as described above and abundance was quantified using the Chromeleon 7.2.10 software with Cobra Wizard defaults. Synthetic standards of 4-phosphoacyl-CoAs could not be synthesized; thus, their elution times were confirmed by injecting reaction mixtures containing LvaE-generated 4-hydroxyacyl-CoAs and ACAD-inactive recombinant ACAD11 (A*). The presence of 4-phosphoacyl-CoA in these reactions was confirmed by LC–MS.

### Assay with free fatty acid substrates using ADP-Glo

The ADP-Glo assay (Promega) was performed according to the manufacturer's instructions with some modifications. Free fatty acid substrates were saponified at 2 M in 20% molar excess NaOH and diluted to 20 mM before use. All solutions were diluted in assay buffer (50 mM HEPES, 1 mM MgCl₂ and 1 mM DTT, pH 7.4). In 96-well plates, 0.5 µM recombinant protein was mixed with 500 µM ATP, 5 µM FAD and 500 µM free fatty acid or LvaE-generated acyl-CoA substrates. Reactions were incubated at 37 °C for 30 min. ADP-Glo reagent (5 µl) was added and incubated with a cover slip for 40 min at room temperature. Kinase detection reagent was then added (10 µl) and incubated for 60 min at room temperature. Luminescence was read using default values on a Biotek Cytation 3 plate reader. An ADP–ATP standard curve was made according to the manufacturer's instructions using reagents supplied in the kit. All conditions were measured in triplicate.

### Acyl-CoA dehydrogenase ferrocenium assay

Ferrocenium assays were conducted as previously described with some modifications[25]. Ferrocenium hexafluorophosphate (Sigma-Aldrich) was dissolved in 10 mM HCl and quantified using Nanodrop pedestal UV–vis spectroscopy. Substrate stock solutions (hexanoyl-CoA (C6:0), decanoyl-CoA (C10:0), myristoyl-CoA (C14:0), stearoyl-CoA (C18:0) and behenoyl-CoA (C22:0); Avanti Polar Lipids) were dissolved in assay buffer (50 mM HEPES, 1 mM MgCl₂ and 1 mM DTT, pH 7.5). Reaction mixtures in assay buffer were prepared in a 96-well plate with 250 µM ferrocenium, 0.5 µM FAD and 250 µM substrate. Each reaction was initiated by adding enzymes (ACAD10 WT, ACAD11 WT, ACADM, ACADS or ACADVL) at a final concentration of 0.5 µM. The decrease in ferrocenium absorbance at 300 nm was measured as a function of time for 5 min using the maximum speed sweep read setting in a Biotek Cytation 3 plate reader. Activity was calculated in units of ΔAbs per min on the basis of the slope of the linear portion of the enzyme kinetic curves,

as determined in Microsoft Excel and GraphPad Prism. All conditions were measured in quadruplicate.

## Enzyme kinetics

For kinase domain activity, kinetic assays were performed using a coupled enzyme system and pure 4-HV-CoA (synthesized by WuXi AppTec) as the substrate. In a half-volume 96-well plate, 50-μl reaction mixtures were prepared with varying concentrations of substrate, excess lactate dehydrogenase (Sigma-Aldrich) and pyruvate kinase (Sigma-Aldrich), 1.5 mM PEP (Sigma-Aldrich), 0.5 mM reduced nicotinamide adenine dinucleotide (Sigma-Aldrich), 1.75 mM ATP, 5 μM FAD and 0.5 μM ACAD11 enzyme. Samples were incubated in the Biotek Cytation 3 plate reader at 37 °C for 10 min. UV–vis measurements were made every ~15 s at 340-nm wavelength. Reaction rates were extracted for each individual reaction by fitting the linear slope to depletion of the signal.

For ACAD domain activity, kinetic assays were performed using malachite green free phosphate quantification and pure 4-PV-CoA (synthesized by WuXi AppTec) as the substrate. The malachite green assay kit (Millipore Sigma) was performed according to the manufacturer's instructions with some modifications. In 96-well PCR-style plates, 20-μl reactions were prepared with 0.5 μM ACAD11, 5 μM FAD and varying concentrations of substrates. Reactions were incubated at 37 °C for 7 min in a heat block with no mixing. Samples were quenched immediately with 20 μl of extraction solvent as described above and further diluted with water to a final volume of 100 μl. In a new transparent 96-well plate, 80 μl of each sample was transferred, along with serial dilutions of a 40 μM free phosphate standard provided in the kit. Malachite green reagent was added (20 μl per well) and incubated for 30 min at room temperature. Absorbance at 620 nm was measured using default values on a Biotek Cytation 3 plate reader. A phosphate standard curve was made in Microsoft Excel according to the manufacturer's instructions and used to quantify phosphate across samples.

Data for both assays were further analyzed and quantified, including Michaelis–Menten curve fitting with standard parameters, in GraphPad Prism. WT reactions were normalized to inactive domain mutant controls to eliminate nonspecific background signal. All conditions were measured in triplicate.

## Cryo-EM sample preparation

For cryo-EM, recombinant ACAD11 (K* and A* mutant) was purified from bacteria as described previously with some modifications. The supernatant from the second Ni-NTA resin incubation step was collected and diluted 20-fold in standard protein buffer without NaCl, resulting in a final NaCl concentration of 15 mM. The diluted solution was subjected to cation-exchange chromatography as described for recombinant ACAD10 above. The eluate from this step was collected, pooled and subjected to size-exclusion chromatography as described previously with some changes. The size-exclusion chromatography buffer used was 50 mM HEPES, 50 mM NaCl, 2% glycerol and 50 μM FAD, pH 7.5. The final sample was also amended to include 100 μM 4-HV-CoA for the kinase-dead mutant or 4-PV-CoA for the ACAD-dead mutant, generated using LvaE and ACAD-dead ACAD11 as described previously. The purest 1.5-ml fraction of eluate was collected and concentrated in an Amicon centrifugal filter before freezing. The final protein concentrations were 230 μg ml⁻¹ for the kinase-dead mutant and 388 μg ml⁻¹ for the ACAD-dead mutant. Cryo-EM samples were prepared on Quantifoil copper 300-mesh 2/2 holey carbon grids and plunge-frozen in a Vitrobot Mark IV (Thermo Fisher Scientific) with a chamber temperature of 4 °C and a humidity set to 95%. Before freezing, grids were freshly plasma-cleaned on a Gatan Solarus 950 plasma cleaner for 60 s. Then, 3 μl of sample was adsorbed onto the grids for 20 s, blotted for 2 s and immediately plunge-frozen in liquid ethane.

## Cryo-EM data processing and map generation

For ACAD11 K* mutant and A* mutant datasets, 2,789 and 2,965 raw videos, respectively, were imported to cryoSPARC (version 3.3.1)[64] and subjected to patch motion correction and contrast transfer function estimation. The high-quality images were manually selected for subsequent data processing. Particles were initially picked using blob picker. Subsequent two-dimensional classes were used as templates for template picker. In total, 722,751 particles of ACAD11 K* mutant and 1,294,186 particles of ACAD11 A* mutant were extracted with a box size of 300 pixels. Classes containing high-resolution particles were selected and used to generate initial models. For the ACAD11 K* mutant, 242,249 final particles were subjected to nonuniform refinement with imposed $D_2$ symmetry. For the ACAD11 A* mutant, heterogenous refinement was conducted to further remove low-quality particles. A total of 297,619 final particles were subjected to local refinement with imposed $D_2$ symmetry. The subsequent two half maps were imported to deepEMhancer (version 0.14)[65]. The maps were postprocessed in high-resolution mode to further improve quality for model building.

## Cryo-EM model building and refinement

For the ACAD11 K* mutant, the crystal structure of human ACAD11 (PDB 2WBI) and the AlphaFold[66] model for ACAD11 were docked into the cryo-EM map and adjusted with rigid-body fit and real-space refinement in Coot (version 0.9.8)[67]. The model was further refined in PHENIX (version 1.20.1)[68]. For the ACAD11 A* mutant, the ACAD11 K* mutant model and the AlphaFold model were docked into the map, followed by rigid-body fit in Coot (version 0.9.8) and real-space refinement in PHENIX (version 1.20.1). Final models were validated using MolProbity[69] and figures were generated with ChimeraX (version 1.5)[70].

## Molecular modeling of ACAD11 A* structure with substrates

Missing residues of the ACAD11 A* structure were first resolved with SwissModel using standard settings. Kinase domain modeling was performed by comparing the kinase structure with choline kinase bound to phosphocholine (PDB 2CKQ)[71]. Core atoms of phosphocholine that would be shared with a 4-HO-CoA substrate for ACAD11 were retained and the molecular structure was extended by adding atoms. To facilitate initial placing of the atoms, a combination of coordinate transfer and manual edits was applied using PyMol (Schrödinger) on a regular computer and HandMol[72] on a virtual reality interface with hand-tracking capabilities. Once a starting position was secured, the system was relaxed through atomistic MD simulations using the CHARMM36m[73] force field in the Gromacs[74] 2022 engine for over 200 ns with restraints on the backbone Cα atoms. Modeling of the ACAD domain was performed by comparing the structure with that reported for an acyl-CoA dehydrogenase with 3-thiooctanoyl-CoA in the active site (PDB 1UDY)[31]. The 4-phospho-octanoyl-CoA substrate was built into the ACAD domain active site by replacing the atoms shared by both substrates (that is, CoA moiety and the first atoms of the carbon chain). The rest of the substrate was modeled by extending the carbon tail toward a cavity using PyMol and HandMol while respecting molecular geometries. Initial model building operations placed the phosphate group nearby, but not in direct contact, with active site residues. MD simulations were performed as described above.

## Conjugation of 4-HDHA to fatty-acid-free BSA

First, 20 mM 4-HDHA (synthesized by WuXi AppTec) in ethanol was dried under nitrogen gas for ~1 h. The dried residues were reconstituted in high-glucose DMEM containing 10% fatty-acid-free BSA (w/v) by sonication in a 37 °C water bath for 5 min using the degas setting, followed by 5 min of rest (four cycles total, inverting tubes between each cycle). To prepare working stocks, conjugated 4-HDHA was diluted tenfold in high-glucose DMEM containing no BSA and sterilized with a 0.22-μm filter. Media were always prepared fresh and used the same day.

## 4-HA catabolism studies in Hepa1-6 cells

Hepa1-6 cells were trypsinized at confluency and counted using the CytoSMART automated cell counter. A total of 1 million cells were seeded in individual 10-cm plates (two plates prepared per sample condition to increase sample amount). After 48 h, cells were rinsed with 1× dPBS (Gibco) and supplemented with high-glucose DMEM containing 10% dialyzed FBS (Bio-Techne, S12850H). For studying 4-HV metabolism, cells were given fresh medium containing 5 mM [$^{13}C_5$]-4-HV (synthesized by WuXi AppTec) or 10 mM unlabeled 4-HV sodium salt (Santa Cruz Biotechnology). Cells were cultured with 4-HV for 6 h before harvest. For studying 4-HDHA metabolism, cells were supplemented with high-glucose DMEM containing 100 µM 4-HDHA conjugated to 1% fatty-acid-free BSA (w/v) (Sigma-Aldrich, A6003). Cells were cultured with 4-HDHA for 24 h before harvest. All conditions were tested as three independent experiments alongside negative controls treated with no exogenous fatty acid.

## Solid-phase extraction of 4-phosphoacyl-CoA and other acyl-CoAs

Acyl-CoAs were extracted from Hepa1-6 cells using adapted solid-phase extraction (SPE) procedures[18,75]. For studies with 4-HV, cells were rinsed twice with cold 1× dPBS and then scraped into 1 ml of cold extraction solvent (methanol and $H_2O$ (1:1, v/v) with 5% glacial acetic acid). After transferring to a 1.5-ml screw-cap microcentrifuge tube, cells were vortexed for 20 s and pelleted (16,300g, 10 min, 4 °C). While spinning, 100-mg 2-(2-pyridyl)-ethyl silica gel cartridges (Supelco) were activated with 1 ml of methanol followed by 1 ml of extraction solvent. Then, 2 ml of total clarified cellular extract from two identical 10-cm plates was passed through the cartridge. The SPE matrix was washed once with 1 ml of methanol with 50 mM ammonium formate (pH 6.3) (1:1, v/v). Compounds were eluted with 0.5 ml of methanol with 50 mM ammonium formate (pH 6.3) (4:1, v/v) twice and 0.5 ml of methanol twice. Next, 2 ml of the total eluate was dried in a speed vacuum concentrator. Dried extracts were stored at −80 °C. Before LC−MS analysis, Hepa1-6 cell acyl-CoA extracts were reconstituted in 50 µl of water and methanol (97:3, v/v) with 10 mM tributylamine (pH 8.2, adjusted with 10 mM acetic acid).

For studies with 4-HDHA, cells were rinsed twice with cold 1× dPBS, scraped into 0.9 ml of cold acetonitrile and IPA (3:1, v/v) and transferred to 2-ml screw-cap tubes. Samples were vortexed for 30 s. Then, 0.3 ml of 0.1 M $KH_2PO_4$ pH 6.7 was added to each sample followed by vortexing for 30 s. The samples were then centrifuged (16,300g, 10 min, 4 °C). Next, 1.2 ml of the supernatants were transferred to new 2-ml screw-cap tubes and acidified with 300 µl of glacial acetic acid followed by vortexing for 15 s. SPE cartridges were preconditioned with 1 ml of methanol followed by 1 ml of acetonitrile, IPA, water and acetic acid (9:3:4:4, v/v/v/v). Then, 1.2 ml of each acidified sample was loaded onto the cartridge (n = 2 technical replicates were loaded sequentially). The cartridges were washed once with 1.2 ml of acetonitrile, IPA, water and acetic acid (9:3:4:4, v/v/v/v). Acyl-CoAs were eluted with 1 ml of methanol with 250 mM ammonium formate (4:1, v/v) three times. Only the second two elutions were collected on the basis of the elution profile of a previous experiment. Then, 2 ml of eluate was transferred to a new 2-ml screw-cap tube and dried by speed vacuum. Dried residues were immediately stored at −80 °C. Before LC−MS analysis, dried extracts were resuspended in 50 µl of 50 mM ammonium acetate and 20% acetonitrile. For data normalization, the residual pellets were dried, boiled in 500 µl of 0.2 M NaOH at 95 °C and assayed for total protein mass with the Pierce BCA Protein Assay Kit (Thermo Scientific).

## Mouse plasma collection and 4-HA extraction

Male and female mice (12–15 weeks old) were fed standard chow (Envigo, TD.00606) or an HFD (Envigo, TD.120528) for 12 weeks. After fasting for 6 h with access to water, mice were anesthetized under isoflurane and euthanized by decapitation. Blood was immediately collected into tubes containing sodium citrate buffer (Covidien). Plasma was isolated by centrifugation (10 min, 2,000g, 4 °C) and the supernatants were transferred to fresh tubes. Plasma samples were flash-frozen in liquid nitrogen and stored at −80 °C until the day of extraction. Plasma 4-HAs were extracted by precipitation in methanol or IPA if fed an HFD. After thawing on ice, 50 µl of plasma was mixed with 1 ml of cold methanol spiked with internal standards. After vortexing for 30 s, samples were placed at −80 °C for 1 h. Samples were centrifuged to pellet insoluble material (16,300g, 10 min, 4 °C). Then, 950 µl of each supernatant was transferred to a new 1.5-ml tube and dried overnight in a speed vacuum concentrator. Dried samples were reconstituted in 50 µl of methanol and water (1:1, v/v) before LC−MS analysis.

## LC−MS of acyl-CoA metabolomics and stable isotope tracing

For in vitro enzyme activity and cell culture experiments, LC−MS analyses of acyl-CoA intermediates derived from 4-HV were conducted using a Vanquish ultra-HPLC (UHPLC) system (Thermo Scientific) coupled to a hybrid quadrupole-Orbitrap MS instrument (Thermo Scientific, Q Exactive) equipped with electrospray ionization (ESI) operating in negative-ion mode. The chromatography was performed at 25 °C using a reverse-phase $C_{18}$ column (2.1 × 100 mm, 1.7-µm particle size; Water, Acquity UHPLC BEH). The chromatography gradient used solvent A (97:3 $H_2O$ and methanol with 10 mM tributylamine adjusted to pH 8.2 using 10 mM acetic acid) and solvent B (100% methanol) as follows: 0–2.5 min, 5% B; 2.5–5 min, linear gradient from 5% B to 20% B; 5–7.5 min, 20% B; 7.5–13 min, linear gradient from 20% B to 55% B; 13–15.5 min, linear gradient from 55%–95% B; 15.5–18.5 min, 95% B, 18.5–19 min, linear gradient from 95% B to 5% B; 19–25 min, 5% B. The flow rate was held constant at 0.2 ml min⁻¹. For the metabolomics method, full MS single-ion monitoring and parallel reaction monitoring (PRM) methods were run with an inclusions list. Eluent from the column was injected into the MS instrument for analysis from 2 min to 17.2 min, at which point flow was redirected to waste for the remainder of the run. Properties for the MS methods included the following: scan range between 500 and 1,000 m/z, automatic control gain target of 5 × 10⁶, maximum injection time of 100 ms, resolution of 70,000 full width at half maximum and isolation windows of 1.4 and 12 m/z for the PRM method for the nonisotopically labeled and isotopically labeled experiments, respectively. Fragmentation of 3-HV-CoA, 4-HV-CoA, 4-PV-CoA and pentenoyl-CoA was achieved by using a normalized collision energy of 25 and a scanning window from 14.5 to 17 min. The autosampler and column compartment were kept at 4 °C and 30 °C, respectively. Data analysis was performed using El-MAVEN (Elucidata)[76] and Xcalibur (Thermo Fisher Scientific) software. Compounds were identified on the basis of retention times matched to pure standards or according to fragmentation patterns reported previously[22].

LC−MS measurement of acyl-CoA intermediates derived from 4-HDHA was performed using a Thermo Vanquish Horizon UHPLC system coupled to a Thermo Exploris 240 Orbitrap MS instrument. For LC separation, a Vanquish binary pump system (Thermo Scientific) was used with a Waters Acquity Premier CSH phenyl-hexyl column (100 mm × 2.1 mm, 1.7-µm particle size) held at 30 °C under 300 µl min⁻¹ flow rate. Mobile phase A consisted of acetonitrile and water (5:95, v/v) with 10 mM ammonium acetate. Mobile phase B consisted of acetonitrile and water (95:5, v/v). For each sample run, mobile phase B was held at 2% for the first 1.5 min and then increased to 15% over the next 1.5 min. Mobile phase B was then further increased to 95% over the following 2.5 min and held for 9 min. The column was then re-equilibrated for 5 min at 2% B before the next injection. Next, 10 µl of sample was injected by a Vanquish Split Sampler HT autosampler (Thermo Scientific) while the autosampler temperature was kept at 4 °C. The samples were ionized by a heated ESI source kept at a vaporizer temperature of 350 °C. Sheath gas was set to 50 units, auxiliary gas was set to 8 units, sweep gas was set to 1 unit and the spray voltage was set to 3,500 V using negative mode. The inlet ion transfer tube temperature was kept at

325 °C with a 70% radiofrequency lens. The identity and retention time of the acyl-CoA derivatives and phosphorylated intermediate were first confirmed from cells fed high-purity 4-HDHA using the PRM method with higher-energy collision dissociation at 30%, targeting select ion fragments generated from the fragmentation of the hydrogen-loss ion. Select fragments were chosen on the basis of previous studies examining the phosphorylated intermediate of shorter-chain 4-hydroxyacyl-CoAs[22]. Quantification of experimental samples was performed using full-scan mode at a resolution of 60,000, targeting the hydrogen-loss ion of 4-hydroxy-DHA-CoA ($m/z = 1,092.3325$) and 4-phospho-DHA-CoA ($m/z = 1,172.2988$). Peak integration was performed using Tracefinder version 5.1 (Thermo Scientific).

For all measurements, raw intensities of verified compound peaks were normalized to total protein mass of the sample and were used as a proxy for compound abundance in cells.

### LC–MS of mouse plasma lipids

Lipid extracts were analyzed using an Agilent Infinity II LC instrument coupled to an Agilent 6495c triple-quadrupole MS instrument as recently described[77]. The LC was equipped with an Agilent RRHD Eclipse Plus C18 column (2.1 × 150 mm, 1.8 µm) with an Agilent Eclipse Plus guard column (2.1 × 5 mm, 1.8 µm) kept at 50 °C. Mobile phase A consisted of 0.1% acetic acid and mobile phase B was acetonitrile and IPA (90:10, v/v). The LC gradient was as follows: 15% B to 33% B at 3.5 min, 38% B at 5.5 min, 42% B at 7 min, 48% B at 9 min, 65% B at 15 min, 75% B at 17 min, 85% B at 18.5 min, 95% B at 19.5 min and 15% B at 21 min, held until 26 min at a constant flow of 0.350 ml min⁻¹. The multisampler was kept at 4 °C and a blank injections were run between every sample. Samples were injected in random order and within 48 h of extraction.

MS analysis was in negative ionization mode with the following parameters: drying gas at 290 °C at 10 L min⁻¹, nebulizer at 35 psi, sheath gas at 350 °C at 11 L min⁻¹, capillary voltage at 3,500 V and nozzle voltage at 1,000 V. A dynamic multiple reaction monitoring (dMRM) method was used for analysis. A library of retention times and transitions was created using synthesized 4-HA standards and other oxygenated lipids from commercial mixes. Additional oxylipins for which standards were not tested were programmed using previously published transitions and retention time correlations[56,78]. Several collision energies between 10 and 40 V were tested to determine transitions for oxylipin standards that lack known transitions or were not detected using previously published transitions. Plasma pooled from all experimental mice that were spiked with all external standards was used to refine and finalize the method for the plasma matrix. Additionally, external standards were injected before and after samples to ensure that there were no retention time shifts or changes in detected analyte abundances because of instrument variability. Retention times and transitions used to identify and quantify targeted molecules, as well as the internal standards used for quantitation, are described in Supplementary Tables 4 and 5.

Data were collected and analyzed using the Agilent MassHunter Suite (version 10.1). Automated peak picking was used to integrate peaks with a signal-to-noise ratio > 3.0 and retention times within 0.30 min of the expected retention time. All peaks were then manually assessed for appropriate shape. Any samples with peak height less than background were removed. Compounds present in at least 60% of samples were reported in the final dataset in units of ng lipid per ml of plasma after normalization to the appropriate internal standard. Data point outliers at least two s.d. outside the group mean were excluded.

### Microscopy of ACAD10/11 organellar localization

COS7 (American Type Culture Collection, CRL-1651) and U-2 OS (American Type Culture Collection, HTB-96) cells were cultured in DMEM containing 10% FBS at 37 °C and 5% CO₂. Cells were seeded on 12-mm (1.5 thickness) coverslips and incubated for 24 h before transfection. Cells were transfected with human full-length ACAD10–FLAG, FLAG–ACAD10, ACAD11–FLAG or FLAG–ACAD11 expressed

under a cytomegalovirus promoter in pcDNA3.1 constructs using Lipofectamine 3000 (Thermo Scientific) in OptiMEM (Gibco). Cells were cultured for 24 h before fixation with 4% paraformaldehyde in OptiMEM for 5 min. Followed by washing with 1× dPBS, cells were permeabilized and blocked in 0.22-µm filter-sterilized 1× blocking buffer (10% normal goat serum (Thermo Scientific, 50197Z), 0.3% Triton X-100 and 0.002% NaN₃) for 1 h at room temperature. Cells were incubated with the following primary antibodies diluted in blocking buffer overnight at 4 °C: mouse anti-FLAG M2 (1:500; Sigma, F1804), rabbit anti-PEX14 (1:500; EMD Millipore, ABC142) and chicken anti-HSP60 (1:500; EnCor Biotechnology, CPCA-HSP60). Samples were washed three times in PBS-T (0.3% Triton X-100) for 10 min each, followed by incubation with secondary antibodies goat anti-mouse IgG (H + L) cross-adsorbed secondary antibody, Alexa Fluor 488 (1:500; Thermo Scientific, A-11001), goat anti-rabbit IgG (H + L) cross-adsorbed secondary antibody, Alexa Fluor 568 (1:500; Thermo Scientific, A-11011) and goat anti-chicken IgY (H + L) cross-adsorbed secondary antibody, Alexa Fluor Plus 647 (1:500; Thermo Scientific, A32933) and counterstained with 1 µg ml⁻¹ DAPI for 1 h at room temperature. Samples were washed three times with PBS-T before mounting on Superfrost Plus slides (Fisher Scientific, 12-550-15) with Fluoromount-G (SouthernBiotech, 0100-01). Samples were imaged on a Zeiss LSM 880 II Airyscan FAST confocal microscope using a ×63 objective and an Airyscan module. ImageJ software (version 2.14.0/1.54f) was used to calculate the pixel intensities of three channels from czi files. Briefly, a 300 × 300 pixel region of interest (ROI) was selected from transiently transfected cells positive for FLAG staining. The intensity for each pixel in the ROI across the three measured channels (488, 568 and 647 nm) was exported in .csv format. These pixel intensities for FLAG–ACAD10 (11 ROIs) or ACAD11–FLAG (15 ROIs) were merged into one dataset. The data were grouped by channel (FLAG, HSP60 and PEX14) and image identifier to normalize the pixel intensity to a range from 0 to 1. For each image, the Pearson correlation coefficient was calculated for the FLAG–HSP60 and FLAG–PEX14 data comparisons. P values for correlation coefficient comparisons were calculated by Wilcoxon rank-sum test using the wilcox.test function from the R package stats (version 4.2.2).

For primary adipocyte imaging, cells were fixed with formalin (Fisher Scientific, MR04586-76) overnight, washed three times with PBS, then incubated with BODIPY 493/503 (Cayman Chemical, 25892) and Hoechst stain (Thermo Scientific, 62249) for 20 min, washed with PBS three times, imaged and quantified at ×20 magnification with an EVOS M5000. Images in Fig. 5k are displayed at ×100 magnification. Analysis of lipid droplet-to-nucleus count ratio was specifically performed using CellProfiler version 4.2.8.

### Primary adipocyte isolation and differentiation

iWAT was excised from female Acad11-KO and WT littermate controls after euthanasia. Excised tissue was cut into fine pieces and incubated in digestion buffer (0.15% w/v collagenase type II (Sigma, C6885) in DMEM (Thermo Scientific, 12491023) with 100 µg ml⁻¹ Primocin (Fisher, NC9392943)) for 1.5 h in a shaking water bath at 37 °C. Excess debris was removed using a 100-µm filter and cells were pelleted with a 500g spin for 5 min. Pellets were washed with DMEM twice and spun down at 500g for 5 min. The pellet was resuspended in 2 ml of DMEM with 10% FBS (Thermo Fisher, 10-437-028) and 100 µg ml⁻¹ Primocin. Cells were immortalized with retroviral expression of SV40 large T antigen (Addgene, 13970) and selected with hygromycin (Thermo Fisher, MT30240CR). For differentiation, cells were grown to confluency in 12-well dishes. At 100% confluency, differentiation medium containing 1 µM dexamethasone, 0.5 mM isobutylmethylxanthine (MP Biomedicals, MP218384891), 5 µg ml⁻¹ insulin (Sigma-Aldrich, I9278-5ML) and 10 nM GW1929 (Fisher Scientific, 16-641-0) was added. After 2 days, differentiation medium was switched to maintenance medium containing 5 µg ml⁻¹ insulin and 10 nM GW1929. Cells were isolated at various time points of differentiation including on days 0, 6 and 8.

## RNA isolation and qPCR analysis

Frozen liver tissue (~50–70 mg) was pulverized with a sterile metal bead in 1 ml of TRIzol reagent (Thermo Fisher) by shaking with the Tissue-Lyser II (Qiagen) (30 Hz, 40 s, three cycles). For cultured cells, 1 ml of TRIzol was added directly to each well and transferred to a 2-ml tube. In between each cycle, samples cooled at 4 °C for 5 min. Samples were transferred to a new tube and mixed with 200 µl of cold chloroform. After inverting ten times, the sample was centrifuged (12,000$g$, 15 min, 4 °C). The upper phase was transferred to a fresh tube containing 500 µl of cold, molecular-grade IPA (Fisher Scientific) and precipitated at −80 °C overnight. After centrifugation (12,000$g$, 15 min, 4 °C), the supernatant was aspirated. The RNA pellet was washed twice with 1 ml of cold 75% ethanol and centrifuged each time (7,500$g$, 2 min, 4 °C). The supernatant was carefully aspirated and RNA pellets were dried in a fume hood. Isolated RNA was dissolved in UltraPure water and quantified by NanoDrop. cDNA was synthesized from 2 µg of RNA using the high-capacity cDNA reverse transcriptase kit (Thermo Scientific). The 20-µl cDNA reactions were diluted 1:10 in UltraPure water. qPCR experiments were prepared in 384-well plates. Diluted cDNA was mixed with target primers and Applied Biosystems PowerUp SYBR green master mix (Thermo Scientific) according to the manufacturer's instructions. Standard curves were generated by pooling cDNA samples. qPCR experiments were performed on the QuantStudio 5 instrument using default settings compatible with SYBR reagent. Gene expression was normalized to *Rps3* abundance.

## Statistics

Statistical analyses were performed in Excel, GraphPad Prism (version 9.4.1 or 10.0.3) or R (version 4.2.2). The in vitro enzyme assay, cell culture data and animal data are expressed as the mean ± s.d. (technical replicates) unless specified otherwise. Primary adipocyte experiment data are expressed as the mean ± s.e.m. (biological replicates). The number of replicates is specified in the figure legends and is represented as individual data points in the panel graphs. Statistical tests used are described in all figure legends. $P$ values are either reported directly in figure panels or are depicted with asterisks (*$P < 0.05$, **$P < 0.01$ and ***$P < 0.001$).

## Reporting summary

Further information on research design is available in the Nature Portfolio Reporting Summary linked to this article.

## Data availability

All enzyme assay and mouse MS spectrometry files were deposited to the MassIVE repository under accession numbers MSV000097576 (in vitro enzyme assay and cell culture metabolomics) and MSV000097543 (mouse plasma lipidomics). Cryo-EM structures are available from the PDB under accession numbers 8V3U and 8V3V. Differential scanning fluorimetry data of purified recombinant proteins and next-generation sequencing of Hepa1-6 CRISPR KO clones are available in Supplementary Information. Other relevant data reported in this work, including raw HPLC data files, are available from the corresponding authors upon reasonable request. Source data, which include raw microscopy and gel images, are provided with this paper.

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

## Acknowledgements

We thank V. Anderson for key discussions and insights on 4-HA metabolism, the J.A.S. lab for their critical assistance with animal experiments and the D.J.P. Lab for their feedback and discussion throughout the duration of this study. We are also grateful to B. Parks for discussions on ACAD11 genetic associations and B. Palfey (University of Michigan) for sharing stocks of 5-deazaFAD. This work was supported by National Institutes of Health awards R01 DK137976 (D.J.P. and J.A.S.), R35 GM131795 and R01 DK098672 (D.J.P.), the Biotechnology Training Program T32 GM008349 (E.H.R. and A.K.B.), the SciMed Advanced Opportunity Fellowship (E.H.R.) and funds from the BJC Investigator Program (D.J.P.). This study leveraged core services at Washington University in St. Louis, including the Genome Engineering and Stem Cell Center (with the guidance of X. Cui) to generate CRISPR–Cas9 KO cells, the Center for Cellular Imaging to perform cryo-EM experiments, the Core Laboratory for Clinical Studies for running mouse plasma chemistry assays and the Center for Drug Discovery (with the guidance of M. Zhou) for assistance with the synthesis strategy for custom compounds used in this study. This work further leveraged University of Wisconsin–Madison facilities, including the Animal Models Core to establish whole-body CRISPR–Cas9 KO mice and the Small Animal Metabolic Phenotyping Facility (with the

guidance of E. Yen) for assistance with nuclear magnetic resonance body composition analysis. J.A.S. is a Freeman Hrabowski Scholar of the Howard Hughes Medical Institute (HHMI). D.J.P. is an investigator of the HHMI. This article is subject to HHMI's Open Access to Publications policy. HHMI lab heads have previously granted a nonexclusive CC BY 4.0 license to the public and a sublicensable license to HHMI in their research articles. Pursuant to those licenses, the author-accepted manuscript of this article can be made freely available under a CC BY 4.0 license immediately upon publication.

## Author contributions

E.H.R., A.K.B. and D.J.P. led the conceptualization, design and execution of this study and wrote this paper. E.H.R. and A.K.B. performed in silico analyses of sequence homology, cloned and purified recombinant enzymes and performed in vitro enzyme assays. A.K.B., J.Z. and P.Y. performed cryo-EM experiments and structural analyses and A.K.B. conducted structure-guided experiments. L.A.A. and M.D.P. performed molecular modeling and simulations. A.J.S. performed cell transfection and immunofluorescence microscopy in vitro experiments. P.F. assisted with ImageJ analysis. E.H.R. performed lipid metabolism studies with cultured cells and extractions for LC–MS analysis. D.B.K., Z.N.B. and D.A.-N. performed LC–MS metabolomics of acyl-CoAs. R.J., G.W. and J.A.S. developed lipidomic methods and measured 4-HAs in vivo. A.C. assisted with literature research and cloning. E.H.R., A.R.C. and J.A.S. maintained mouse colonies and performed mouse experiments. J.A.S. performed primary adipocyte ex vivo differentiation studies. T.C. and B.F.P. provided consultation and assisted with cloning and purifying LvaE.

## Competing interests

The authors declare no competing interests.

## Additional information

**Extended data** is available for this paper at https://doi.org/10.1038/s41594-025-01596-4.

**Correspondence and requests for materials** should be addressed to Judith A. Simcox or David J. Pagliarini.

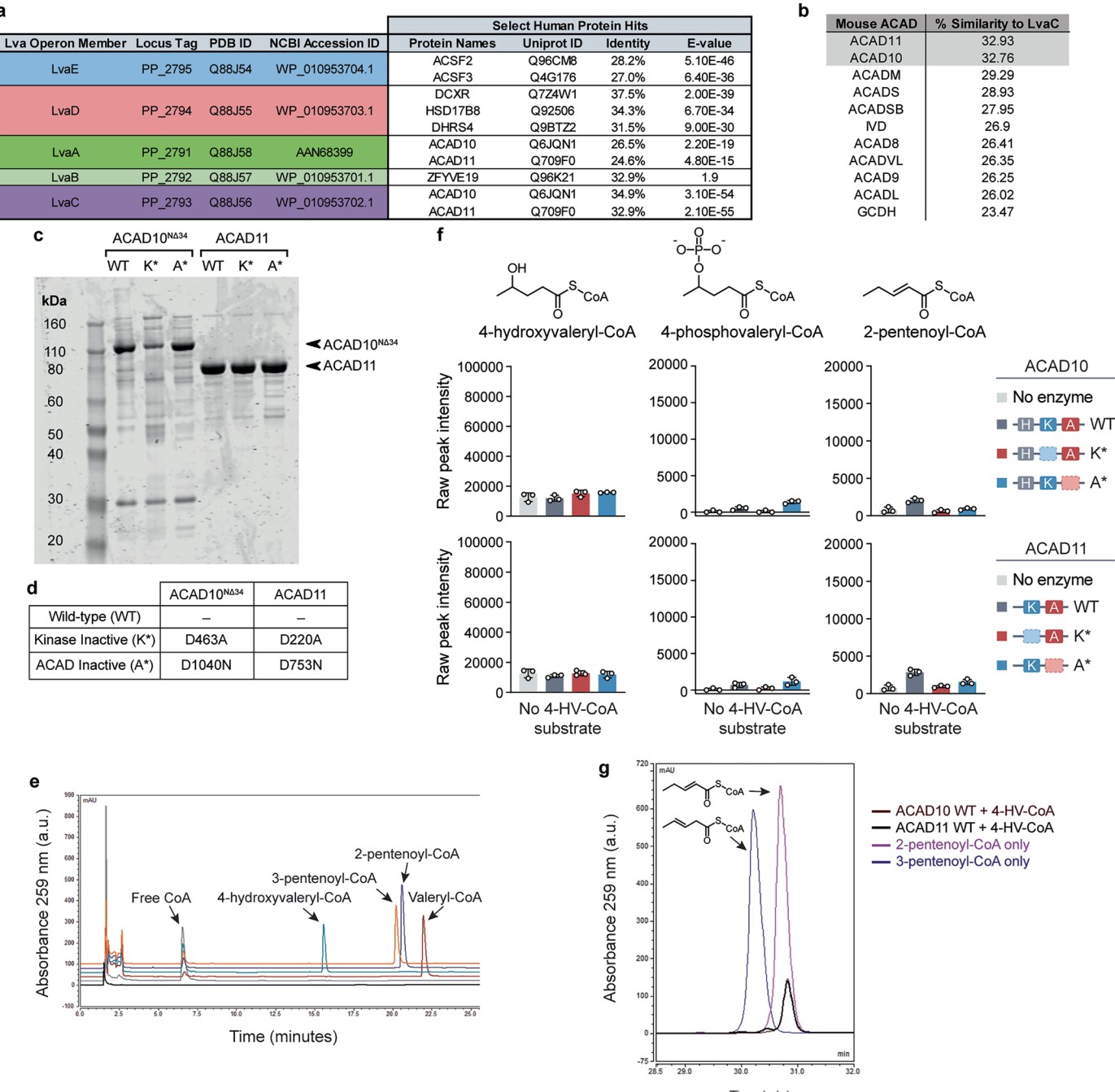

**Extended Data Fig. 1 | ACAD10/11 process 4-HA substrates *in vitro*.**
**a**, Sequence identity of *lva* operon enzymes to human proteins as reported by UniProtKB database. **b**, Sequence similarity of LvaC relative to mouse ACAD family according to pairwise alignments. **c**, SDS-PAGE gel of recombinant mouse ACAD10$^{N\Delta34}$ and ACAD11 protein preparations from *P. pastoris* or RIPL *E. coli* orthogonal expression systems, respectively. WT = wild type; K* = Kinase inactive mutant; A* = ACAD inactive mutant. **d**, Table describing the site-specific mutations to inactivate recombinant ACAD10$^{N\Delta34}$ and ACAD11 kinase and ACAD catalytic domains. **e**, HPLC chromatogram of acyl-CoA standards. **f**, LC-MS analysis of *in vitro* enzyme reactions in Fig. 1c lacking 4-HV-CoA substrate.

Displayed are the raw signal intensities and chemical structures of 4-HV-CoA, 4-PV-CoA, and 2-pentenoyl-CoA (above corresponding graphs). Top and bottom bar graphs represent data from reactions with recombinant ACAD10$^{N\Delta34}$ and ACAD11, respectively. The same "no enzyme" control data is displayed on both sets of graphs for clearer comparison. Data are expressed as mean -/+ SD (*n* = 3 technical replicates). **g**, Representative HPLC trace of pentenoyl-CoA isomers formed from 500 µM 4-HV-CoA by recombinant wild-type ACAD10$^{N\Delta34}$ and ACAD11 *in vitro* (brown and black traces, respectively) in comparison to LvaE-generated pentenoyl-CoA standards (dark purple and blue traces, respectively).

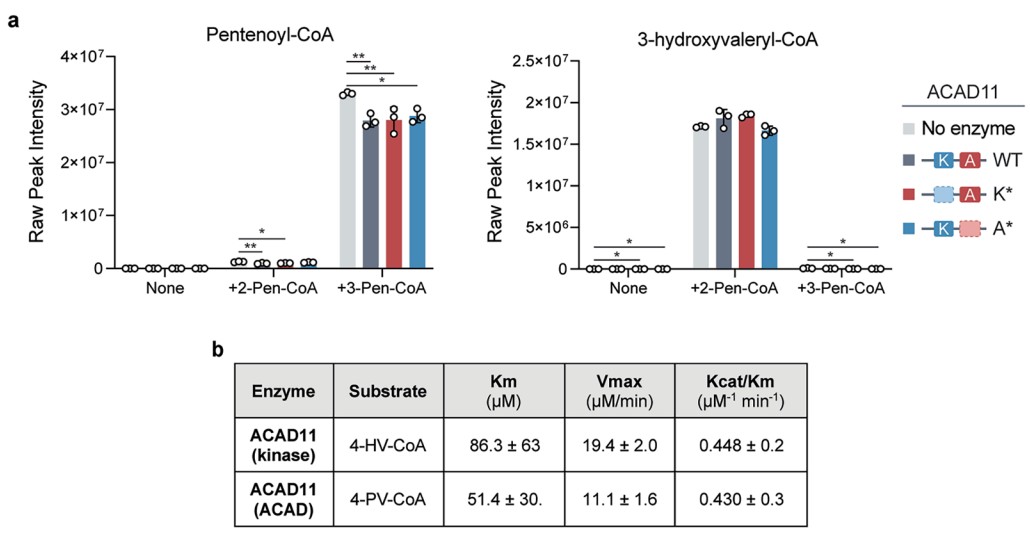

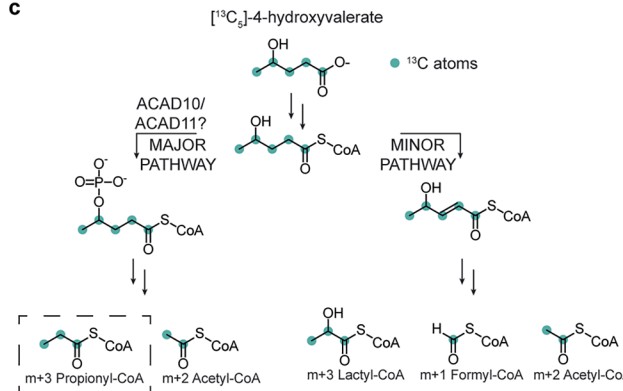

**Extended Data Fig. 2 | ACAD10/11 activity kinetics and their metabolism of 4-HA substrates in cells through the major pathway. a**, Assay testing the non-specific hydration activity in recombinant ACAD11 reactions. 2-pentenoyl-CoA (2-Pen-CoA) and 3-pentenoyl-CoA (3-Pen-CoA) were tested as substrates. Raw signal intensities of peaks corresponding to substrates and 3-hydroxyvaleryl-CoA product are depicted. Data represents mean -/+ SD (*n* = 3 technical replicates). **b**, Table of calculated Km, Vmax, and Kcat/Km values for ACAD11 kinase and ACAD domain activities. Values were obtained by fitting to

Michaelis-Menten curves in Figs. 1e and 1f. Data are expressed as mean -/+ 95% symmetric confidence interval. **c**, Expected labeling pattern of acyl-CoA catabolites derived from the catabolism of [$^{13}C_5$]-4-hydroxyvalerate (4-HV) by the "major" or "minor" pathways. M + 3 propionyl-CoA labeling reflects catabolism through the 4-phosphovaleryl-CoA intermediate, presumably catalyzed by ACAD10/11. Statistics: One-way ANOVA with Dunnett's multiple comparisons test comparing all conditions to the respective "no enzyme" control. *P*-values are represented as asterisks: * = *P* < 0.05, ** = *P* < 0.01, *** = *P* < 0.001.

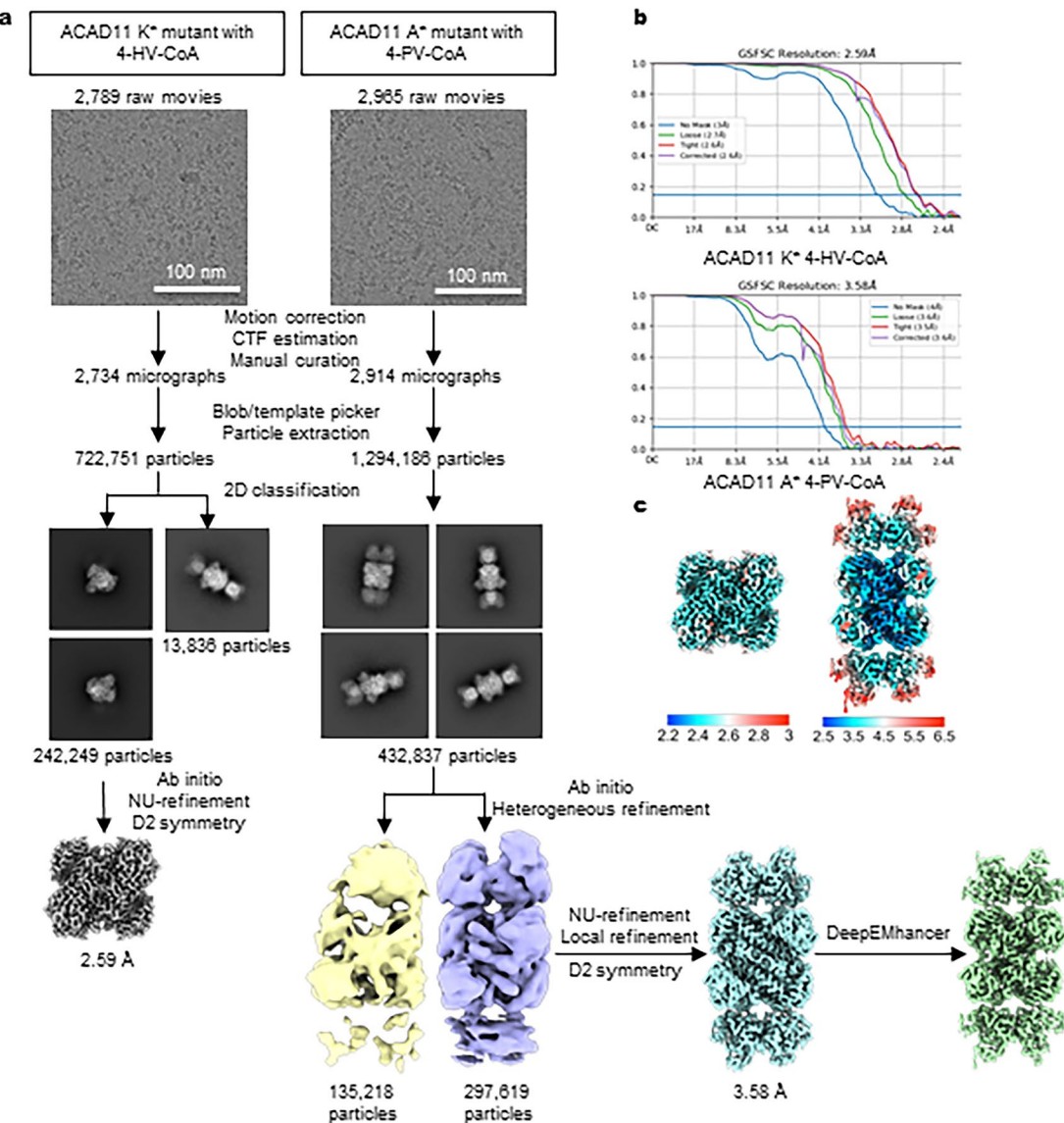

**Extended Data Fig. 3 | Cryo-EM analysis of the ACAD11 K* and A* mutants. a**, Data processing flowcharts. **b**, FSC curves of non-uniform refinement of the ACAD11 K* mutant and local refinement of the ACAD11 A* mutant. **c**, Local resolution of the final maps.

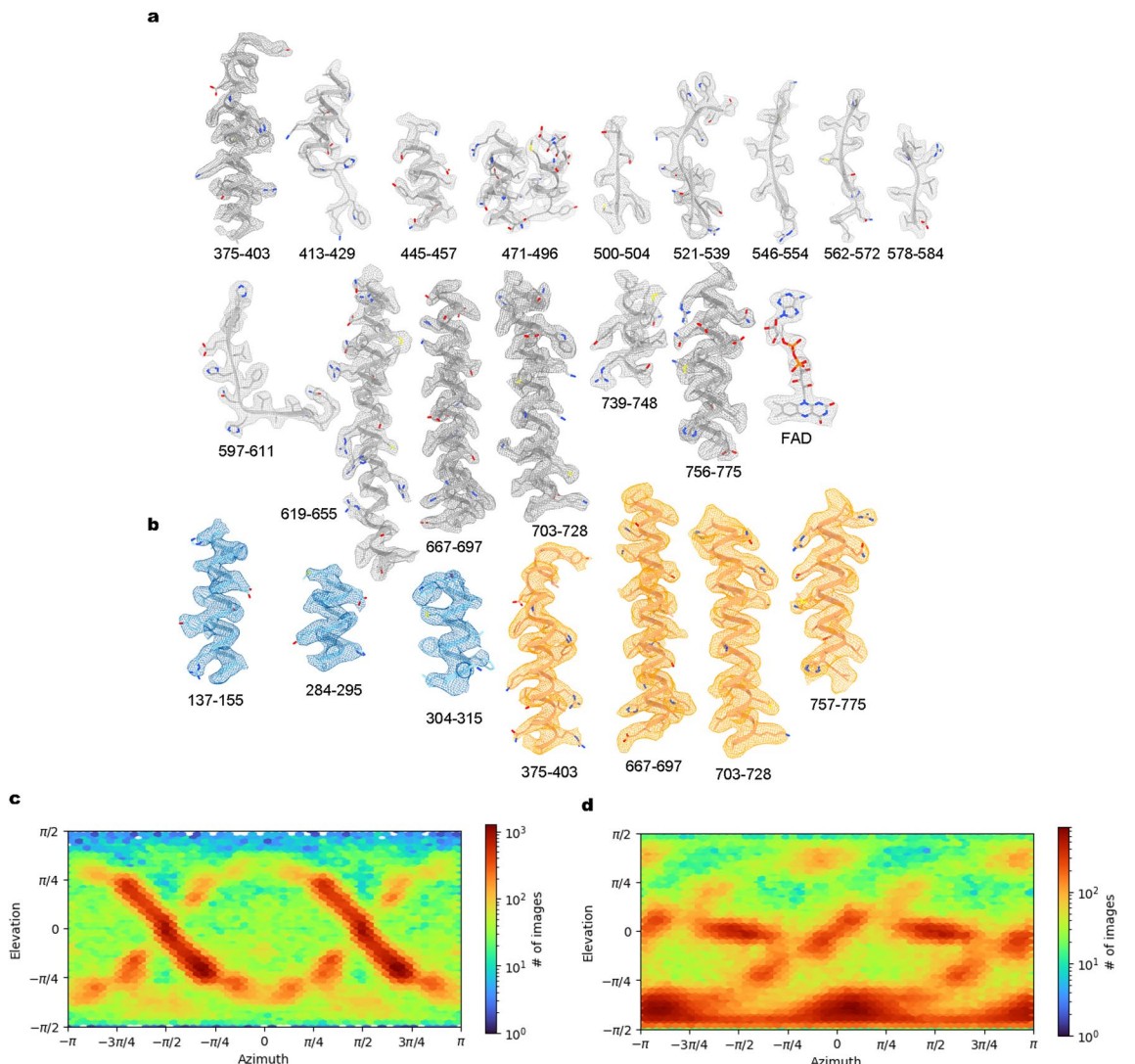

**Extended Data Fig. 4 | Cryo-EM densities of ACAD11 K\* and A\* mutants. a, b**, Representative cryo-EM densities of the ACAD11 K\* mutant (**a**) and A\* mutant (**b**). **c, d**, Angular distribution plots of ACAD11 K\* mutant (**c**) and A\* mutant (**d**).

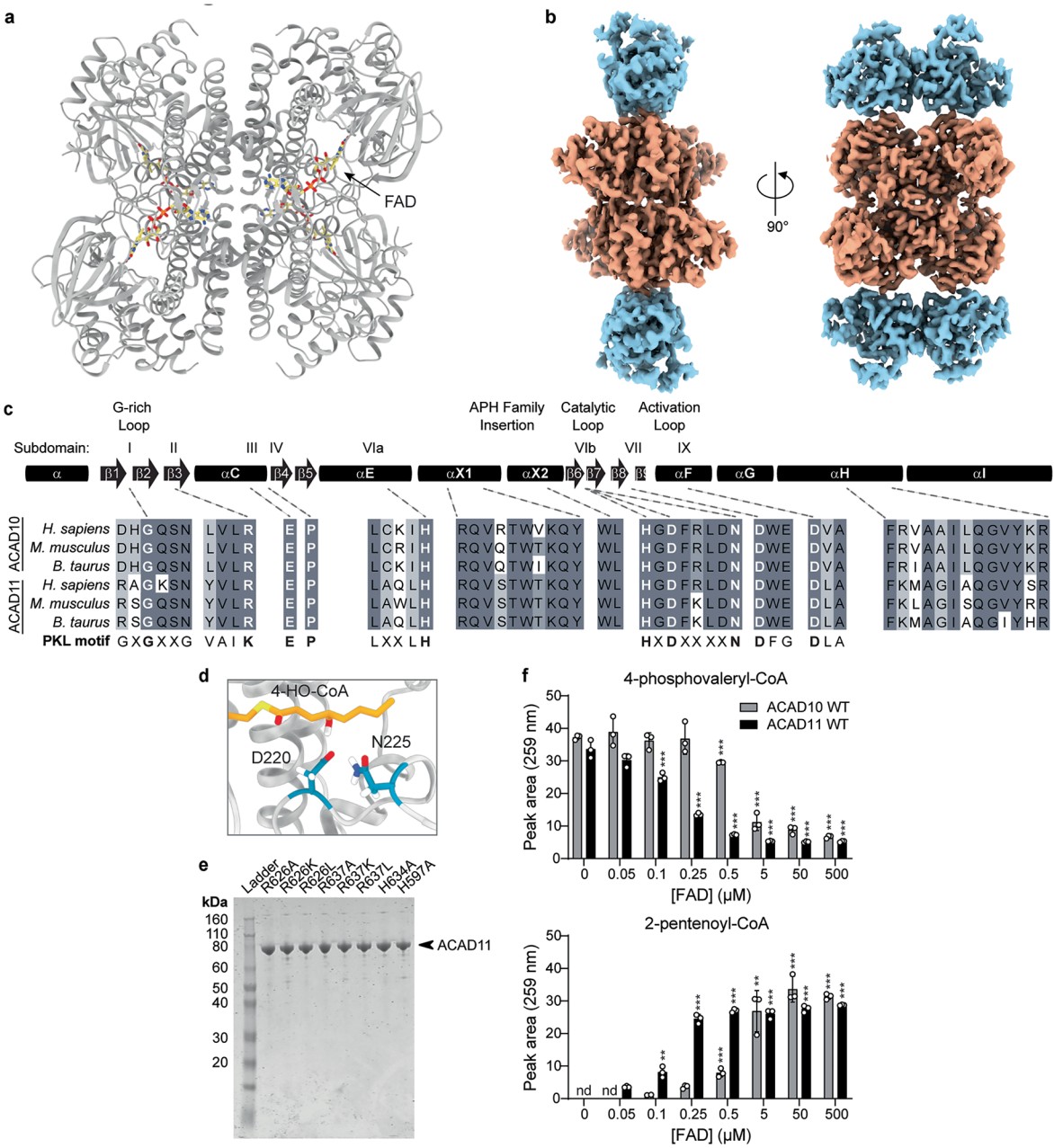

**Extended Data Fig. 5 | Cryo-EM of ACAD11 reveals structural determinants of molecular function. a**, Cryo-EM structure of the ACAD11 K* mutant incubated with 4-HV-CoA. FAD molecules are highlighted in yellow. **b**, Front and side angle view of full-length ACAD11 A* mutant cryo-EM structure when incubated with 4-PV-CoA. **c**, Sequence alignment of key regions of the kinase domain. The overall canonical PKL subdomain structure is shown at the top. ACAD10/11 homologs from *H. sapiens*, *M. musculus*, and *B. taurus* are compared to PKL consensus motifs, with key conserved residues highlighted. **d**, MD modeling of 4-hydroxyoctanoyl-CoA (4-HO-CoA; orange sticks) in the kinase domain relative

to the catalytic D220 and a highly conserved N225 (blue sticks). **e**, Coomassie-stained gels of recombinant ACAD11 ACAD-domain mutants purified from *E. coli*. **f**, UV-Vis HPLC quantification (259 nm) of 4-PV-CoA and 2-pentenoyl-CoA in reactions containing wild-type ACAD10/11 and varying concentrations of FAD (mean -/+ SD; *n* = 3 technical replicates). Statistics: Two-way ANOVA with Šídák's multiple comparisons test comparing each condition to the respective 0 μM FAD reaction control. *P*-values are represented as asterisks: * = *P* < 0.05, ** = *P* < 0.01, *** = *P* < 0.001. nd = not detected.

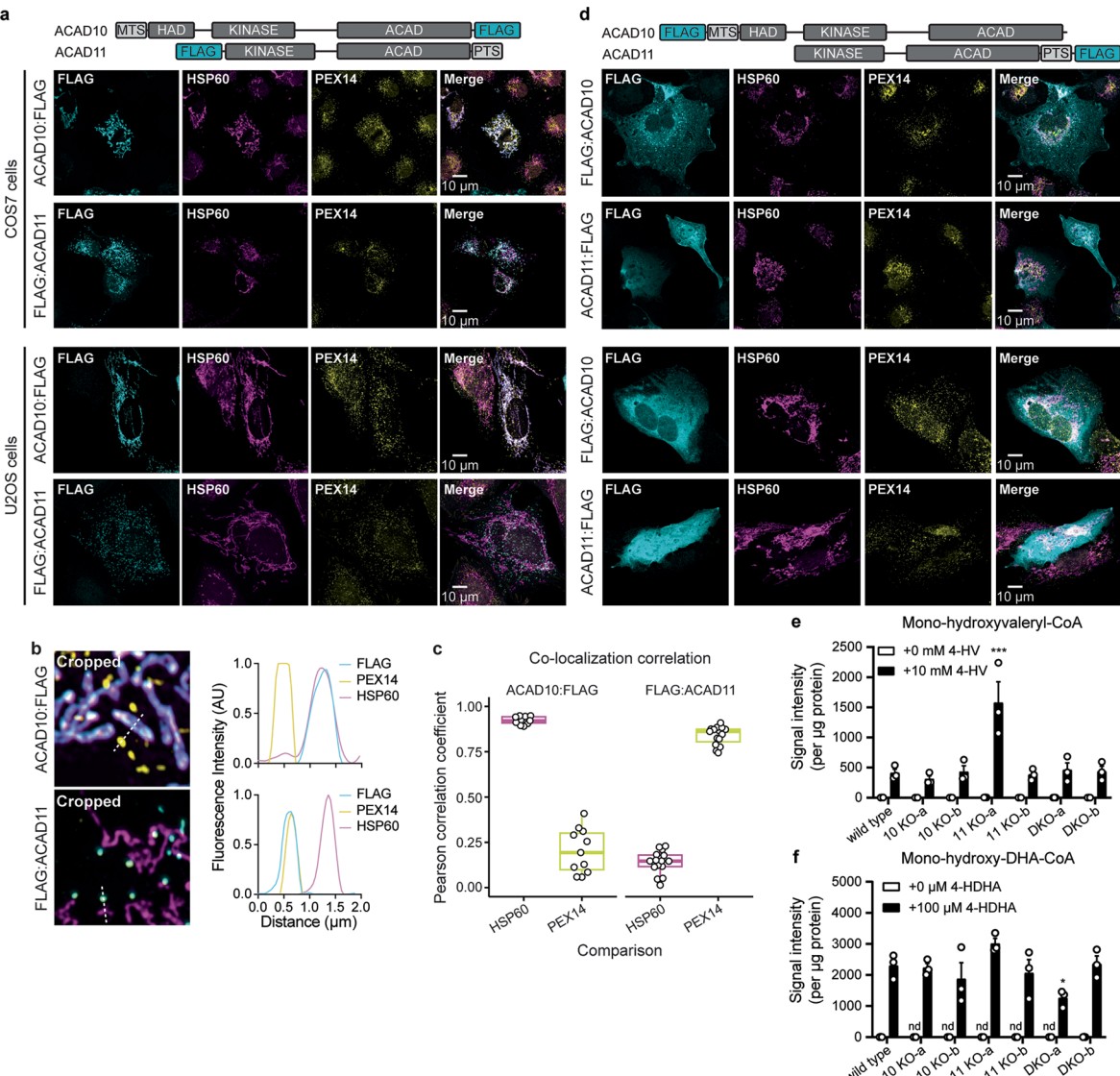

**Extended Data Fig. 6 | ACAD10/11 mediate catabolism of distinct 4-HAs in different organelles. a**, Immunofluorescence imaging of overexpressed FLAG-tagged constructs in COS7 and U2OS cells at 10 µm scale. **b**, Crop of merged images depicted in Fig. 4a and fluorescence emission line quantification (right graphs). Localization overlap of ACAD10:FLAG (cyan; top row) and FLAG:ACAD11 (cyan; bottom row) are compared to mitochondrial HSP60 (magenta) and peroxisomal PEX14 (yellow). Scale bar = 2 µm. **c**, Pearson correlation coefficients of the overlap between ACAD10:FLAG or FLAG:ACAD11, HSP60, and PEX14 across n = 11 and n = 15 technical replicate regions of interest (ROIs), respectively (ROIs are defined as 300 ×300 pixel areas; each ROI captures a portion of one transiently transfected COS7 cell). Each data point represents one independent

ROI. Data are shown as box-plots. Centerline, median; box limits, $25^{th}$ to $75^{th}$ percentiles; whiskers, minimum and maximum points. **d**, Immunofluorescence imaging of overexpressed constructs with FLAG-tags swapped to opposite termini in COS7 and U2OS cells at 10 µm scale. **e, f**, Signal intensity of peaks that likely correspond to 4-hydroxyacyl-CoAs of exogenously delivered 4-HAs. Data are represented as the mean signal intensity (normalized to total protein) -/+ SD ($n$ = 3 technical replicates). Statistics: Two-sided Wilcoxon Ranked Sum Test was used to obtain Pearson correlation coefficients (**c**), two-way ANOVA with Šídák's multiple comparisons test comparing each treatment group to the respective "wild type" control (**e, f**). $P$-values are represented as asterisks: * = $P$ < 0.05. nd = not detected.

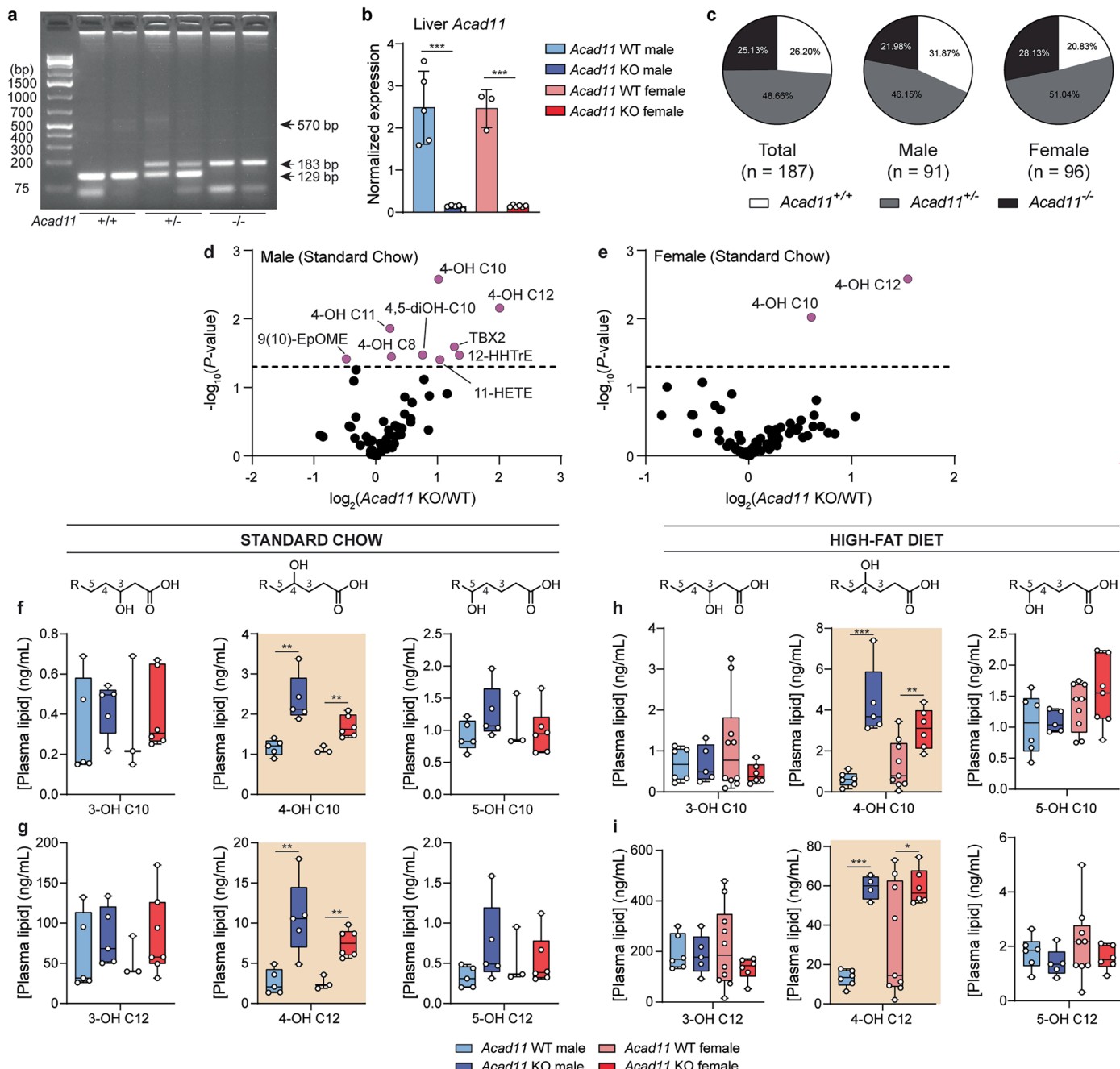

**Extended Data Fig. 7 | Mice lacking ACAD11 have perturbed 4-HA homeostasis.**
**a**, Genotyping of *Acad11* KO mouse colony. **b**, Normalized mRNA expression of *Acad11* in liver tissue. Data are normalized to *Rps3* levels and expressed as mean -/+ SD. **c**, Mendelian birth ratios of *Acad11* mice. **d, e**, Volcano plots of log$_2$-transformed fold changes of 4-HAs and other hydroxylated lipids in mice fed standard chow relative to statistical significance. Male (**d**) and female (**e**) *Acad11* KO mice were compared to respective littermate controls. **f-i**, Concentrations of 4-OH C10 (top) and 4-OH C12 (bottom) in comparison with their respective 3-OH and 5-OH fatty acid isomers in plasma across all genotypes and sex. Data are

expressed as box-plots. Centerline, median; box limits, 25$^{th}$ to 75$^{th}$ percentiles; whiskers, minimum and maximum points. Mice fed a defined standard chow for 5 days (**f, g**). Mice fed a HFD for 12 weeks (**h, i**). For **b** and **d-g**: Male WT $n = 5$, Male KO $n = 5$; Female WT $n = 3$, Female KO $n = 6$. Mice were approximately 4-5 months old when sacrificed. For **h** and **i**: Male WT $n = 6$, Male KO $n = 5$; Female WT $n = 10$, Female KO $n = 7$. Mice were approximately 3 months old at the start of HFD feeding. Statistics: Two-sided Student's *t*-test comparing *Acad11* KO mice to respective littermate controls (**b** and **f-i**), two-sided multiple *t*-tests (**d, e**). *P*-values are represented as asterisks: * = $P < 0.05$, ** = $P < 0.01$, *** = $P < 0.001$.

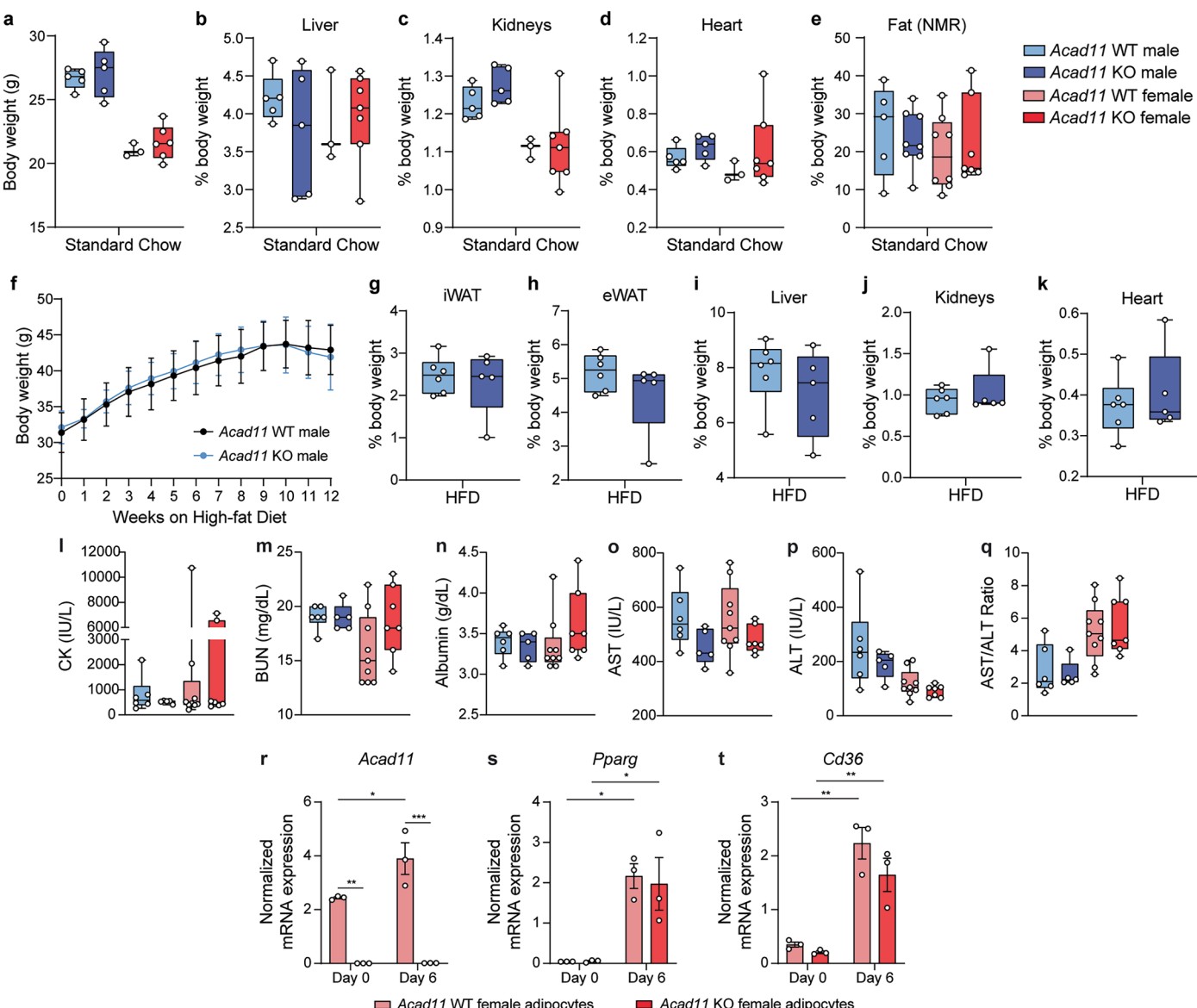

**Extended Data Fig. 8 | Physiological measurements of *Acad11* mice on standard chow and high-fat diet. a-d,** Total body (**a**), liver (**b**), kidneys (**c**), and heart (**d**) weights of WT and *Acad11* mice fed standard chow. Data are shown as box-plots (Male WT *n* = 5, Male KO *n* = 5; Female WT *n* = 3, Female KO *n* = 6). Mice were approximately 4-5 months old when sacrificed. **e**, NMR body composition analysis quantifying total fat mass. Data are shown as box-plots (Male WT *n* = 5, Male KO *n* = 8; Female WT *n* = 8, Female KO *n* = 7). Standard chow-fed mice were approximately 8 months old when sacrificed. **f-k**, Body weight over time (mean -/+ SD) (**f**), iWAT (**g**), eWAT (**h**), liver (**i**), kidneys (**j**), heart (**k**) weights of male WT and *Acad11* KO mice after 12 weeks of HFD feeding. **l-q**, Plasma markers of liver and kidney pathology and function in HFD-fed WT and *Acad11* KO mice. For **f-q**, data are expressed as box plots. Male WT *n* = 6, Male KO *n* = 5; Female WT *n* = 9,

Female KO *n* = 7. One WT is excluded from weight analyses due to renal agenesis. Mice were approximately 3 months old at the start of HFD feeding. For box-plots: centerline, median; box limits, 25th to 75th percentiles; whiskers, minimum and maximum points. **r-t**, Normalized gene expression in female WT and *Acad11* KO primary adipocytes at zero and six days of differentiation. *Acad11* (**r**), *Pparg* (**s**), and *Cd36* (**t**) expression were normalized to *Rps3* levels. Data are expressed as mean -/+ SEM (*n* = 3 independent biological replicates). Statistics: Two-sided Student's *t*-test comparing *Acad11* KO mice to respective WT littermate controls (**a-e** and **g-q**), two-way repeated measures ANOVA with Šídák's multiple comparisons test comparing *Acad11* KO mice to wild-type mice at each time point (**f**), and two-way ANOVA with Šídák's multiple comparisons test (**r-t**). *P*-values are represented as asterisks: * = *P* < 0.05, ** = *P* < 0.01, *** = *P* < 0.001.

## Reporting Summary

## Statistics

For all statistical analyses, confirm that the following items are present in the figure legend, table legend, main text, or Methods section.

| n/a | Confirmed | |
|---|---|---|
| ☐ | ☒ | The exact sample size (*n*) for each experimental group/condition, given as a discrete number and unit of measurement |
| ☐ | ☒ | A statement on whether measurements were taken from distinct samples or whether the same sample was measured repeatedly |
| ☐ | ☒ | The statistical test(s) used AND whether they are one- or two-sided<br>*Only common tests should be described solely by name; describe more complex techniques in the Methods section.* |
| ☐ | ☒ | A description of all covariates tested |
| ☒ | ☐ | A description of any assumptions or corrections, such as tests of normality and adjustment for multiple comparisons |
| ☐ | ☒ | A full description of the statistical parameters including central tendency (e.g. means) or other basic estimates (e.g. regression coefficient) AND variation (e.g. standard deviation) or associated estimates of uncertainty (e.g. confidence intervals) |
| ☐ | ☒ | For null hypothesis testing, the test statistic (e.g. *F*, *t*, *r*) with confidence intervals, effect sizes, degrees of freedom and *P* value noted<br>*Give P values as exact values whenever suitable.* |
| ☒ | ☐ | For Bayesian analysis, information on the choice of priors and Markov chain Monte Carlo settings |
| ☒ | ☐ | For hierarchical and complex designs, identification of the appropriate level for tests and full reporting of outcomes |
| ☐ | ☒ | Estimates of effect sizes (e.g. Cohen's *d*, Pearson's *r*), indicating how they were calculated |

*Our web collection on statistics for biologists contains articles on many of the points above.*

## Software and code

Policy information about availability of computer code

| Data collection | Electrophoresis gel images and Western blots were imaged using LI-COR Image Studio (v5.2.5). HPLC data was collected using Thermo Scientific Chromeleon console (v7.2.10). Plate reader data was collected using BioTek Gen5 software (v1.11.5). All LC-MS data was collected using the following commercial software: Thermo Scientific Xcalibur (v4.3 and v4.6), Agilent MassHunter Acquisition Method (v10.1) |
|---|---|
| Data analysis | Cryo-EM data was processed using cryoSPARC (v3.3.1), deepEMhancer (v0.14), COOT (v0.9.8), PHENIX (v1.20.1), and ChimeraX (v1.5). Molecular modeling simulations were performed using Gromacs (2022), VMD (1.9.4). Manual analysis of models was performed using PyMOL (v3.1.4) via the HandMOL VR interface. HPLC data was analyzed using Thermo Scientific Chromeleon console (v7.2.10) and Graphpad Prism (v10.0.3). Plate reader and other in vitro assay data was analyzed using Microsoft Excel or Graphpad Prism (v9.4.1 or v10.0.3), including all statistical analyses unless otherwise mentioned. Metabolomics and stable isotope tracing mass spectrometry data was processed in El-MAVEN (v0.12.0), Thermo Fisher Scientific Xcalibur (v4.3 and v4.6), and Thermo Scientific Tracefinder (v5.1) softwares. Mouse plasma mass spectrometry data was analyzed using the Agilent MassHunter Suite (v10.1). Fluorescence intensity profiles for microscopy were prepared using ImageJ software (v2.9.0/1.53t) and/or CellProfiler (v4.2.8). Quantitative PCR data was analyzed using Applied Biosystems QuantStudio Real-Time PCR software (v1.5.1). Organellar localization Pearson correlations were analyzed using R (v4.2.2). |

For manuscripts utilizing custom algorithms or software that are central to the research but not yet described in published literature, software must be made available to editors and reviewers. We strongly encourage code deposition in a community repository (e.g. GitHub). See the Nature Portfolio guidelines for submitting code & software for further information.

## Data

Policy information about availability of data

All manuscripts must include a data availability statement. This statement should provide the following information, where applicable:
- Accession codes, unique identifiers, or web links for publicly available datasets
- A description of any restrictions on data availability
- For clinical datasets or third party data, please ensure that the statement adheres to our policy

All enzyme assay, mouse physiology, microscopy, and mass spectrometry datasets used to generate figures are available in the Source Data files. Protein sequences used for mammalian homology analysis were obtained from the UniProt database (Taxon ID 40674). Raw mass spectrometry files were deposited to the MassIVE repository under the accession numbers MSV000097576 (in vitro enzyme assay and cell culture metabolomics) and MSV000097543 (mouse plasma lipidomics). The accession numbers for cryo-electron microscopy structures are PDB 8V3U and 8V3V. Raw microscopy and gel images are provided as Supplementary Information. Other relevant data reported in this work, including raw HPLC data files, are available from the corresponding author upon reasonable request.

## Research involving human participants, their data, or biological material

Policy information about studies with human participants or human data. See also policy information about sex, gender (identity/presentation), and sexual orientation and race, ethnicity and racism.

| | |
|---|---|
| Reporting on sex and gender | This study does not involve human participants or use human data. |
| Reporting on race, ethnicity, or other socially relevant groupings | This study does not involve human participants or use human data. |
| Population characteristics | This study does not involve human participants or use human data. |
| Recruitment | This study does not involve human participants or use human data. |
| Ethics oversight | This study does not involve human participants or use human data. |

Note that full information on the approval of the study protocol must also be provided in the manuscript.

# Field-specific reporting

Please select the one below that is the best fit for your research. If you are not sure, read the appropriate sections before making your selection.

☒ Life sciences    ☐ Behavioural & social sciences    ☐ Ecological, evolutionary & environmental sciences

For a reference copy of the document with all sections, see nature.com/documents/nr-reporting-summary-flat.pdf

# Life sciences study design

All studies must disclose on these points even when the disclosure is negative.

| | |
|---|---|
| Sample size | All experiments were performed in triplicate or more. No statistical approaches were used to determine sample size. Sample sizes chosen (3 or more samples) were rational as they allowed for appropriate statistical testing. |
| Data exclusions | Lipid measurements that were poorly detected (not detected in >40% samples or signal below background levels) and outlier lipid measurements (two standard deviations outside of the group mean) were excluded from final lipidomics analysis. |
| Replication | Three or more technical or biological replicates were used in each experiment when applicable. All attempts at replicating experimental results were successful. |
| Randomization | Measurements were made quantitatively and automated when possible to mitigate Investigator bias. When necessary, injection order of HPLC and MS samples was randomized to limit bias or noise introduced by technical factors (i.e., running all technical or biological replicates of a given condition consecutively). |
| Blinding | For animal work, Investigators were only blinded to genotype of mice during mouse euthanization, sample collection, and subsequent sample analysis. For in vitro enzymology and cell culture work, blinding was not possible since the experimental setup requires knowledge of the sample identities. |

# Reporting for specific materials, systems and methods

We require information from authors about some types of materials, experimental systems and methods used in many studies. Here, indicate whether each material, system or method listed is relevant to your study. If you are not sure if a list item applies to your research, read the appropriate section before selecting a response.

## Materials & experimental systems

| n/a | Involved in the study |
|-----|----------------------|
| ☐ | ☒ Antibodies |
| ☐ | ☒ Eukaryotic cell lines |
| ☒ | ☐ Palaeontology and archaeology |
| ☐ | ☒ Animals and other organisms |
| ☒ | ☐ Clinical data |
| ☒ | ☐ Dual use research of concern |
| ☒ | ☐ Plants |

## Methods

| n/a | Involved in the study |
|-----|----------------------|
| ☒ | ☐ ChIP-seq |
| ☒ | ☐ Flow cytometry |
| ☒ | ☐ MRI-based neuroimaging |

## Antibodies

| | |
|---|---|
| Antibodies used | Western blot Primary antibody: Rabbit anti-GFP (1:1000; Abcam; ab6556)<br><br>Western blot Secondary antibody: IRDye 680RD Goat anti-Rabbit IgG SeconWdary Antibody (1:5000; Licor Bio; 926-68071)<br><br>Imaging Primary antibodies: Mouse anti-FLAG M2 (Sigma; F1804); rabbit anti-PEX14 (1:500; EMD Millipore; ABC142); chicken anti-HSP60 (1:500; EnCor Biotechnology; CPCA-HSP60).<br><br>Imaging Secondary antibodies: Goat anti-Mouse IgG (H+L) Cross-Adsorbed Secondary Antibody, Alexa Fluor 488 (1:500; Thermo Scientific; A-11001); Goat anti-Rabbit IgG (H+L) Cross-Adsorbed Secondary Antibody, Alexa Fluor 568 (1:500; Thermo Scientific; A-11011); Goat anti-Chicken IgY (H+L) Cross-Adsorbed Secondary Antibody, Alexa Fluor Plus 647 (1:500; Thermo Scientific; A32933) |
| Validation | The rabbit anti-GFP (Abcam; ab6556) was validated by the manufacturer according to their website. Antibody was determined to be specific to all variants of Aequorea victoria GFP (UniProt ID: P42212; Molecular weight: 27kDa).<br><br>The mouse anti-FLAG M2 (Sigma; F1804) was validated by the manufacturer according to their website. Specificity was determined by detection of target protein on a Western blot from an E. coli, plant, or mammalian crude cell lysate. Sensitivity was determined by dot blot, detecting as little as 2 ng of target protein.<br><br>The rabbit anti-PEX14 (EMD Millipore; ABC142) was validated by the manufacturer according to their website. Specificity was determined by detection of target protein on a Western blot from NIH/3T3 cell lysate and human liver lysate.<br><br>The chicken anti-HSP60 (EnCor Biotechnology; CPCA-HSP60) was validated by the manufacturer according to their website. Specificity was determined by detection of target protein on a Western blot from SH-SY5Y and HeLa cells. |

## Eukaryotic cell lines

Policy information about cell lines and Sex and Gender in Research

| | |
|---|---|
| Cell line source(s) | Hepa1-6 cells (CRL-1830), U-2 OS cells (HTB-96), and COS7 cells (CRL-1651) were purchased from the American Type Culture Collection (ATCC). Hepa1-6 parental cells were used to generate single and double knockouts of ACAD10 and/or ACAD11 via CRISPR/Cas9 technology (performed by the Genome Engineering and Stem Cell Center (GESC) at Washington University in St. Louis). |
| Authentication | Hepa1-6 KO cell lines were authenticated by GESC by STR analysis and next-generation sequencing of amplicons of the targeted regions. After transfection of sgRNAs and clonal selection, KO clones were chosen for this study if all indels at the select target sites were predicted to introduce premature stop codons and no wild-type alleles were detected. Next-generation sequencing results for all KO lines are included in the Supplementary Information file. |
| Mycoplasma contamination | All cell lines used in this study tested negative for mycoplasma contamination. |
| Commonly misidentified lines (See ICLAC register) | No commonly misidentified lines were utilized in this study. |

## Animals and other research organisms

Policy information about studies involving animals; ARRIVE guidelines recommended for reporting animal research, and Sex and Gender in Research

| | |
|---|---|
| Laboratory animals | Animal work was conducted with cohorts of mice in the C57BL/6NJ mixed background. In vivo experimental groups were 3 months old, 4-5 months old, or 8 months old and included n = 3-10 mice. All experimental mice were evaluated with litter-mate controls. |
| Wild animals | This study did not utilize wild animals. |

| Reporting on sex | Both male and female mice were evaluated in all in vivo experiments. |
|---|---|
| Field-collected samples | This study did not utilize field-collected samples. |
| Ethics oversight | All animal experiments were approved by the Institutional Animal Care and Use Committee of the College of Agricultural and Life Sciences at the University of Wisconsin-Madison. |

Note that full information on the approval of the study protocol must also be provided in the manuscript.

## Plants

| Seed stocks | Plant models were not used in this study |
|---|---|
| Novel plant genotypes | n/a |
| Authentication | n/a |

