## [Peer Review File · Nature Structural & Molecular Biology]

ACAD10 and ACAD11 enable mammalian 4-hydroxy acid lipid catabolism

Corresponding Author: Professor David Pagliarini

Version 0:

Decision Letter:

Our ref: NSMB-A50472-T

13th Mar 2025

Dear Dr. Pagliarini,

Thank you for submitting your revised manuscript "ACAD10 and ACAD11 enable mammalian 4-hydroxy acid lipid catabolism" (NSMB-A50472-T). It has now been seen by the original referee and their comments are below. As you know, we asked you to focus on responses to technical concerns of reviewers #3 and #4. However, only reviewer #3 was available to assess this revision, and we asked them to comment on your responses to reviewer #4 as well. The reviewer find that the paper has improved in revision, and therefore we'll be happy in principle to publish it in Nature Structural & Molecular Biology, pending minor revisions to satisfy the referees' final requests and to comply with our editorial and formatting guidelines. Please note that out of the final reviewer comments, we would only expect you to provide magnified images in the final revision.

We are now performing detailed checks on your paper and will send you a checklist detailing our editorial and formatting requirements in about 2-3 weeks. Please do not upload the final materials and make any revisions until you receive this additional information from us.

Sincerely,
Kat

Katarzyna Ciazynska, PhD
(she/her)
Senior Editor
Nature Structural & Molecular Biology
<https://orcid.org/0000-0002-9899-2428>

Reviewer #3 (Remarks to the Author):

The authors have generally addressed all my comments making the manuscript and the statistics in it stronger and suitable for publications.

I have two more comments that should be addressed and should not require a lot of work:

-For the bodipy imaging, magnified crops should be provided for each wider field as the differences cannot be spotted at such magnification; thus, the size/number of lipid droplets cannot be judged.

-As many physiological studies in the mice were considered a but superficial, I would request that the authors make the MRI scans to assess body composition and calculate their fat and lean mass. Since HA in tissues cannot be easily measured, giving this analyses would supply at least a demonstration of a broad impact of the mutation in this biochemical pathway on the whole animal.

Referees' comments:

Referee #3 (Remarks to the Author):

The authors have generally addressed all my comments making the manuscript and the statistics in it stronger and suitable for publications. I have two more comments that should be addressed and should not require a lot of work:

-For the bodipy imaging, magnified crops should be provided for each wider field as the differences cannot be spotted at such magnification; thus, the size/number of lipid droplets cannot be judged.

Thank you for pointing this out. We have increased the magnification of these images in Figure 5k by 5-fold to ensure that lipid droplet size/count differences are visually clearer (see below).

-As many physiological studies in the mice were considered a bit superficial, I would request that the authors make the MRI scans to assess body composition and calculate their fat and lean mass. Since HA in tissues cannot be easily measured, giving this analyses would supply at least a demonstration of a broad impact of the mutation in this biochemical pathway on the whole animal.

Thank you for this suggestion. We agree that MRI scans would be powerful in revealing the total fat and lean mass differences between WT and ACAD11 KO mice on high-fat diet. In addition to measuring the weights of two major WAT depots, we broadened our study of the physiological impact of ACAD11 genetic deficiency by also measuring the percent body weight of liver, kidney, and heart (Fig. 5h-j), all of which are tissues that have strong ACAD11 expression. Although these tissues only represent a portion of the total fat and lean mass compositions, we observed specific weight differences in the kidneys and hearts of ACAD11 KO mice which would not be clear from MRI scan alone. We did not focus on skeletal muscle, since ACAD11 is poorly expressed in this tissue (GTEx Portal). However, the effect of ACAD11 deficiency on muscle physiology is an important question that requires further investigation and would benefit strongly from the use of MRI.